# Activation of the insulin receptor by an insulin mimetic peptide

Junhee Park [1,6], Jie Li [2,6], John P. Mayer[3], Kerri A. Ball[3], Jiayi Wu [1], Catherine Hall [1], Domenico Accili[4], Michael H. B. Stowell [3]✉, Xiao-chen Bai [2,5]✉ & Eunhee Choi [1]✉

Insulin receptor (IR) signaling defects cause a variety of metabolic diseases including diabetes. Moreover, inherited mutations of the IR cause severe insulin resistance, leading to early morbidity and mortality with limited therapeutic options. A previously reported selective IR agonist without sequence homology to insulin, S597, activates IR and mimics insulin's action on glycemic control. To elucidate the mechanism of IR activation by S597, we determine cryo-EM structures of the mouse IR/S597 complex. Unlike the compact T-shaped active IR resulting from the binding of four insulins to two distinct sites, two S597 molecules induce and stabilize an extended T-shaped IR through the simultaneous binding to both the L1 domain of one protomer and the FnIII-1 domain of another. Importantly, S597 fully activates IR mutants that disrupt insulin binding or destabilize the insulin-induced compact T-shape, thus eliciting insulin-like signaling. S597 also selectively activates IR signaling among different tissues and triggers IR endocytosis in the liver. Overall, our structural and functional studies guide future efforts to develop insulin mimetics targeting insulin resistance caused by defects in insulin binding and stabilization of insulin-activated state of IR, demonstrating the potential of structure-based drug design for insulin-resistant diseases.

Insulin binds and activates a receptor tyrosine kinase, the insulin receptor (IR), at the cell surface[1–3]. The activated IR triggers signaling cascades to control many facets of physiology. Dysfunctional IR signaling causes metabolic disorders such as diabetes, a heterogenous disease whose treatment poses a major challenge. The rising prevalence of diabetes and the limitations of established therapeutic options mandate the development of additional strategies to complement existing approaches. Moreover, genetic mutations of IR cause rare untreatable severe insulin resistance syndromes such as Donohue syndrome and Rabson-Mendenhall syndrome[4–12]. While insulin, insulin sensitizers, leptin and insulin-like growth factor 1 (IGF1) can

temporarily achieve glycemic control and alleviate the symptoms of these complications[13–16], these approaches do not restore normal IR signaling. Further, there is no specific long-term treatment for patients with severe insulin resistance diseases where their life expectancy is only a few years.

In contrast to other receptor tyrosine kinases, IR is folded and assembled as a stable, disulfide-linked dimer in the absence of insulin binding[17,18] (Fig. S1a). The primary insulin binding site (Site-1) is composed of the L1 domain and the C-terminal region of IRα (α-CT)[19–21]. The secondary insulin binding site (Site-2) is located solely on the FnIII-1 domain[20]. We and others have recently demonstrated that the

[1]Department of Pathology and Cell Biology, Vagelos College of Physicians and Surgeons, Columbia University, New York, NY 10032, USA. [2]Department of Biophysics, University of Texas Southwestern Medical Center, Dallas, TX 75390, USA. [3]Department of Molecular, Cellular & Developmental Biology, University of Colorado, Boulder, CO 80309, USA. [4]Department of Medicine, Vagelos College of Physicians and Surgeons, Columbia University, New York, NY 10032, USA. [5]Department of Cell Biology, University of Texas Southwestern Medical Center, Dallas, TX 75390, USA. [6]These authors contributed equally: Junhee Park, Jie Li. ✉e-mail: stowellm@colorado.edu; Xiaochen.Bai@UTSouthwestern.edu; EC3477@cumc.columbia.edu

binding of multiple insulin monomers to both site-1 and site-2 synergistically induces complex conformational changes of IR from the Λ-shaped apo-IR dimer to a T-shaped symmetric IR dimer with full activity[20,22–29]. Inhibition of insulin binding to IR site-2 leads to an asymmetric conformation and only partial activation of IR[24].

Previous phage display affinity screening identified peptides that can bind to the IR with sub-micromolar affinity[30–33]. Strikingly, while lacking sequence homology to insulin, these peptides induce robust IR auto-phosphorylation and potently activate IR signaling. One of these insulin mimetic peptides, S597, lowers blood glucose and increases de novo lipogenesis in the liver and deposition of triglycerides in adipose tissue of Zucker diabetic fatty rats[34], suggesting that S597 could activate insulin-like signaling in vivo. Furthermore, S597 protects against atherosclerosis associated with metabolic syndromes by reducing leukocytosis[35]. Notably, S597 activates the AKT pathways as effectively as insulin but is less efficient in activating the MAPK pathways in blood leukocytes and skeletal muscles[32,35]. These results suggest that S597 engages the IR differently from native insulin, inducing a distinctive (unique) conformation of the IR during receptor activation. S597 is predicted to have two structural components (previously referred to as Site-1 and Site-2) that simultaneously bind two distinct regions on the IR[32]. To distinguish these two components from the insulin binding site-1 and site-2 of IR, we termed the Site-1 and Site-2 of S597 as S597-component-1 and S597-component-2, respectively. The structure of the L1 domain of IR in complex with the S597-component-1 alone has been determined by X-ray crystallography[36], illustrating how the S597-component-1 engages the IR. However, due to a lack of structural information, how the S597 agonist binds to IR and induces IR activation remains largely unclear.

Here, we determine the cryo-EM structure of full-length IR bound with the S597-component-2 alone at a resolution of 3.5 Å, showing that S597-component-2 is folded as a short α-helix and binds to the side-surface of the FnIII-1 domain of IR. We next solve the cryo-EM structure of the IR in complex with the full-length S597 at a resolution of 5.4 Å. Unlike insulin, which binds to two distinct sites on the IR at a maximal 4:1 stoichiometry (insulin: IR dimer), in each half of the IR/S597 complex, one S597 concurrently binds to both the L1 domain of one protomer and the FnIII-1 domain of another, resulting a 2:1 stoichiometry (S597: IR dimer). The binding of two S597 molecules to the IR induces and stabilizes an extended T-shaped conformation that allows the intracellular kinases to undergo efficient auto-phosphorylation. Strikingly, our structural and functional results demonstrate that, because S597 binds and activates IR in a unique manner, S597 fully activates IR mutants that disrupt insulin binding or destabilize the insulin-induced compact T-shape. Collectively, this work elucidates the S597-induced activation mechanism for the IR and suggests that insulin mimetics can restore normal IR signaling in severe insulin resistance diseases. More importantly, these data imply that such peptide mimetics may be viable therapeutics for treating these fatal congenital diseases.

## Results

### The structure of full-length IR bound with the S597-component-2

The binding pattern of S597-component-1 on the IR has been previously characterized by X-ray crystallography[36]. In order to explore how the S597-component-2 engages the IR, we synthesized the N-terminal 20 residues of S597 comprising only component-2 and the linker (S597-N20, Fig. 1a and Fig. S1b) and determined the cryo-EM structure of full-length mouse IR/S597-component-2 complex at overall 3.6 Å resolution (Fig. 1b and Figs. S2–S4). The resolution for one-half of the IR/S597-component-2 complex was further improved to 3.5 Å resolution after symmetry expansion and focused refinement. Structural comparison between S597-component-2 bound IR and apo-IR revealed no major structural differences (Fig. 1c), suggesting that

the S597-component-2 alone cannot induce the conformational change of IR required for signaling.

Additional strong densities were identified at the side-surface of the FnIII-1 domain of each IR protomer that could be unambiguously assigned to the S597-component-2 (Fig. 1d, e). Fourteen residues that constitute the S597-component-2 could be built into a cryo-EM map based on clear side-chain densities (Fig. 1f, g). The model shows that S597-component-2, assuming a short α-helical conformation, packs tightly against the side-surface of the FnIII-1 domain of IR mainly through hydrophobic interactions. Particularly, Leu2, Trp6, Ile9, and Tyr14 of S597-component-2 positioned along one side of the helix make strong interactions with several hydrophobic residues in the FnIII-1 domain of IR, including Leu486, Pro537, Pro549, Gly550, and Leu552. In addition, Glu5 of S597-component-2 and Arg16 of the linker region form salt bridges with Arg479 and Asp535 of the IR FnIII-1 domain, respectively, further contributing to the strong IR/S597-component-2 interaction. It is noteworthy that these key residues are conserved between human and mouse IR (Fig. S5).

### Both components-1 and -2 of S597 are required for the IR activation

It has been previously shown that the S597-component-1 binds the L1 domain of IR in a similar fashion as the α-CT motif[36] (Fig. 2b–d). Therefore, the binding of S597-component-1 to the L1 domain of IR would disrupt the association between the L1 domain and α-CT motif, which together serve as the insulin binding site-1. Our structure shows that the S597-component-2 binds to a similar surface on the FnIII-1 domain of IR as that used for the site-2 insulin binding (Fig. 2e, f). Thus, the IR binding sites of S597-components-1 and -2 largely overlap with the insulin binding sites-1 and -2 on IR, respectively. Because the binding of multiple insulins to both site-1 and site−2 is required for optimal IR activation, we hypothesize that both the binding of S597-component-1 to the IR L1 domain and the binding of S597-component-2 to the IR FnIII-1 domain are required for the S597-induced IR activation.

To test this hypothesis, we introduced F64A or F96A mutations into IR to disrupt the IR L1/S597-component-1 interaction. Because we used the A isoform of the IR for structure determination, and because there was no significant difference between the A and B isoforms of IR activation upon stimulation with S597 (Fig. S6a, b), mutations were introduced into the A isoform of the IR. 293FT cells expressing the IR F96A mutant exhibited great deficiency in S597-dependent IR activation (Fig. 2g, h). However, the IR F64A mutant could still be activated by S597, suggesting that the substitution of Phe64 with a small hydrophobic residue is not sufficient to disrupt the IR L1/S597 interaction (Fig. 2g, h). The Phe64 and Phe96 of the L1 domain of IR form hydrophobic interactions with Phe705 and Phe701 in α-CT that are critical for insulin site-1 binding[20,21,37] (Fig. 2c). As expected, both IR F64A and F96A mutants showed defective insulin-dependent activation (Fig. 2g, h). These results confirm the importance of IR/S597-component-1 interaction in IR activation and also indicate that the L1/α-CT interaction, which is critical for insulin-dependent IR activation, is not required for S597-dependent IR activation (Fig. 2g, h).

We next examined the action of S597 on the activation of an IR K484E/L552A double mutant[20], which was expected to weaken both the IR/site-2 insulin and the IR/S597-component-2 interactions (Fig. 2g, h). Consistent with our structural models, the IR K484E/L552A mutant could not be activated efficiently by either insulin or S597. Previous studies have shown that S597 does not activate the insulin-like growth factor 1 receptor (IGF1R)[32,35], another receptor tyrosine kinase closely related to IR. Some key residues that are critical for IR/S597-component-2 interaction, including Tyr477, Arg479, Lys484, Arg488, Trp551, and Arg554, are not conserved in the IGF1R (Fig. 1g and Fig. S5)[38,39], partially explaining the specificity of S597 on IR activation.

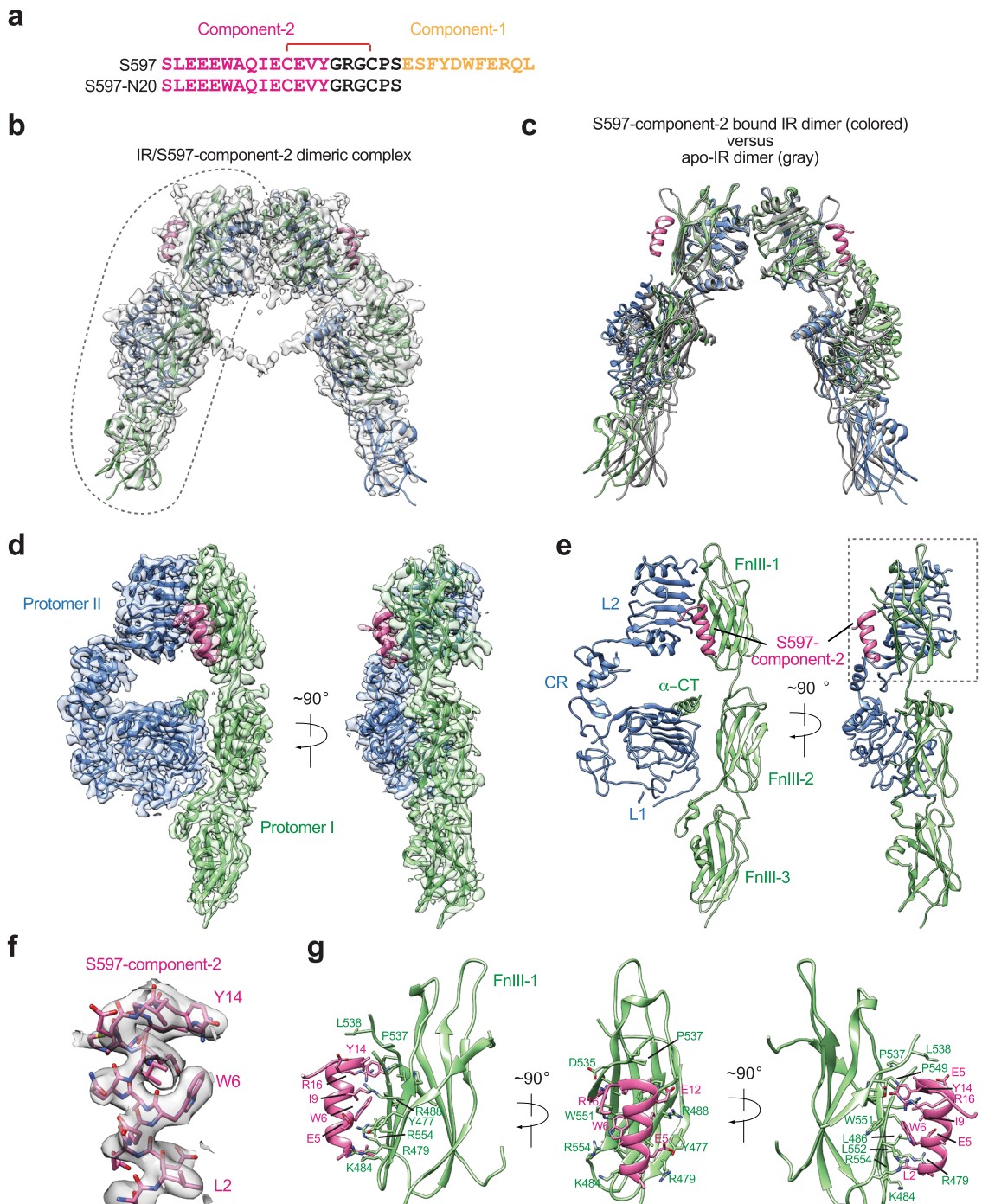

**Fig. 1 | Overall structure of the IR/S597-component-2 complex. a** Sequences of S597 and S597-N20. The residues in S597-component-1 are marked in yellow, S597-component-2 in pink, and linker in black. The disulfide bond was indicated as red. **b** The 3D reconstruction of the IR dimer with two S597-N20 peptides bound. **c** The ribbon representation of the IR dimer with two S597-N20 peptides bound fitted into the cryo-EM map at 3.6 Å resolution. **d** 3D reconstructions of the IR dimer from the gray dashed line in **b** after forced 3D refinement. **e** The ribbon representations of the IR dimer from the gray dashed line in **b** after forced 3D refinement fitted into the cryo-EM map at 3.5 Å resolution. **f** Close-up view of the S597-N20. **g** Close-up view of the binding of S597-N20 (pink) at the FnIII-1 domain of IR (green).

To further investigate the function and mechanism of S597 binding on IR activation, we designed and synthesized S597 with a mutation of the residue that is essential for binding to either L1 or FnIII-1 domains. Specifically, we introduced an S597-W6A mutation to disrupt the IR FnIII-1/S597-component-2 interaction and an S597-F27A mutation to interrupt the IR L1/S597-component-1 interaction (Fig. 2a and Fig. S1b). Consistent with the results of IR mutants (Fig. 2g, h), S597-W6A and -F27A mutants, as well as S597-N20, showed greatly reduced potency in triggering IR activation relative to S597 and insulin

(Fig. 2i, j). These data validate the functional importance of both L1/S597 and FnIII-1/S597 interactions for IR activation.

## Structure of the full-length IR/S597 complex
To reveal the molecular mechanism that underlies S597-induced IR activation, we determined the cryo-EM structure of full-length IR in complex using the complete S597 sequence comprising both components-1 and −2 as well as the linker connecting these two components (Figs. S4, S7). The cryo-EM map of the IR/S597 complex

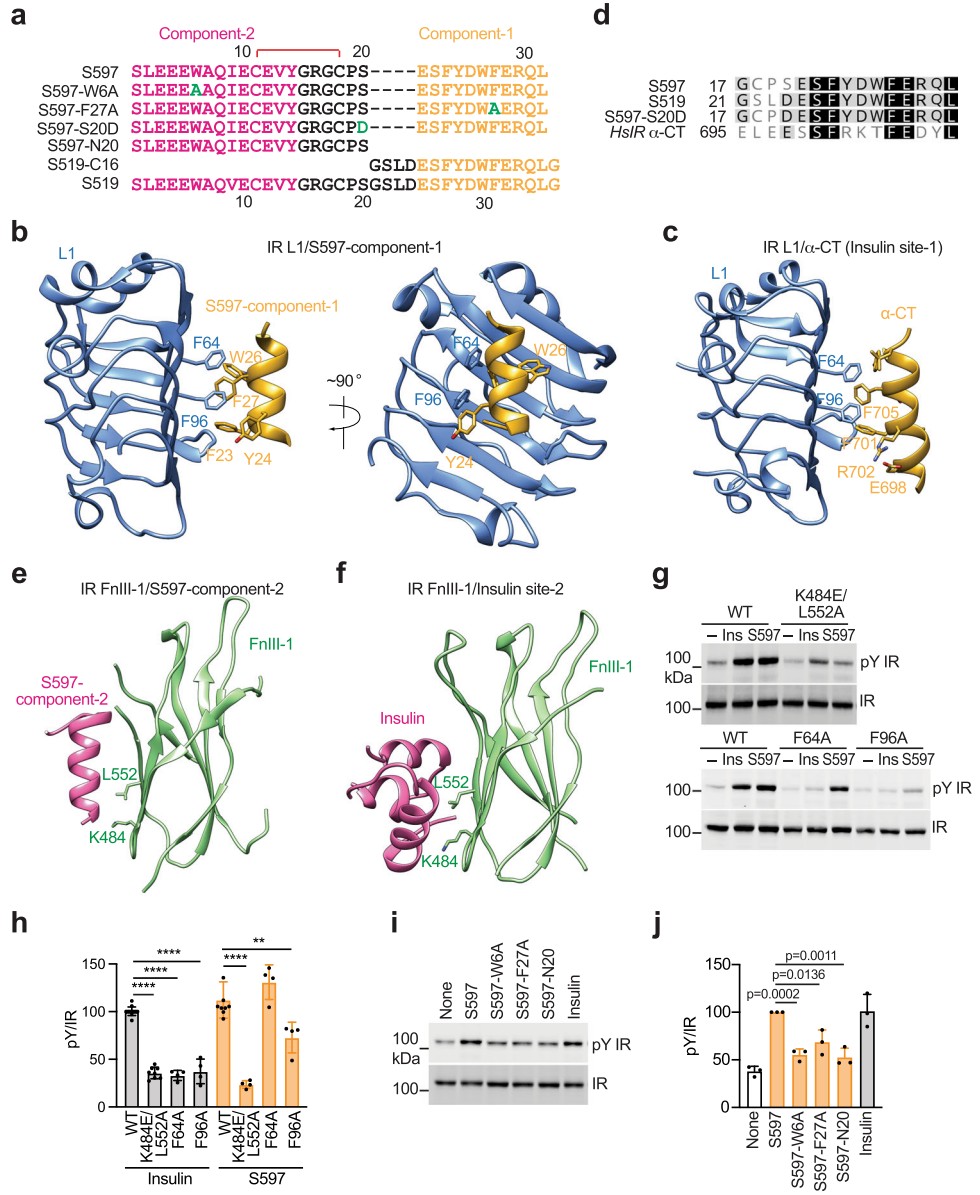

**Fig. 2 | S597 binds to the L1 and FnIII-1 domains of IR. a** Sequences of S597, S597-W6A, S597-F27A, S597-S20D, S597-N20, S519-C16, and S519. The residues in component-1 are marked in yellow, component-2 in pink, mutation in green, and linker in black. The disulfide bond was indicated as red. **b** Close-up view of the binding of S597-component-1 (yellow) at the L1 domain of IR (blue). PDB: 5J3H. **c** Close-up view of the binding of insulin (yellow) at the L1/α-CT domains of IR (blue). PDB: 6PXV. **d** Sequence alignment of human (Hs) IR, S597, S519, and S597-S20D. **e** Close-up view of the binding of S597-component-2 (pink) at the FnIII-1 domain of IR (green). **f** Close-up view of the binding of insulin (pink) at the FnIII-1 domain of IR (green). PDB: 6PXV. **g** Auto-phosphorylation of IR (pY IR) by 10 nM insulin or S597 for 10 min in 293FT cells expressing wild-type (WT) IR or the indicated IR mutants. **h** Quantification of the western blot data shown in **g**. Mean ± SD. Levels of pY IR were normalized to total IR levels and shown as intensities relative to that of IR WT in insulin-treated cells. Each experiment was repeated four times. Significance calculated using two-tailed Student's *t*-test; between WT and mutants, **$p < 0.01$ and ****$p < 0.0001$. The exact $p$ values are provided in the source data. **i** Auto-phosphorylation of IR by 10 nM insulin or S597 analogs for 10 min in 293FT cells expressing IR WT. **j** Quantification of the western blot data shown in **i**. Mean ± SD. Levels of pY IR were normalized to total IR levels and shown as intensities relative to that of IR WT in S597-treated cells. Each experiment was repeated three times. Significance was calculated using a two-tailed Student's *t*-test. Source data are provided as a Source Data file.

was determined at 5.4 Å resolution, indicating the structural flexibility of this complex. Nonetheless, we were able to build a model for the entire complex by rigid-body fitting the structures of the S597-component-1 bound L1 domain (determined previously by X-ray crystallography[36]), the S597-component−2 bound FnIII-1 domain (determined in this work), as well as the other domains of IR (determined previously by cryo-EM[20]) without major structural adjustments.

In contrast to the IR/S597-component-2 complex, the structure of IR/S597 assumes an extended T-shaped symmetric conformation, bringing together the two membrane-proximal stalks (Fig. 3a, b). This

structural feature suggests that IR bound with the full-length S597 represents the active ligand-receptor complex. Two S597 molecules were observed in the top region of the extended T-shaped IR. Each S597, which adopts a helix-loop-helix fold, is embraced between two IR protomers at the top part of the extended T. The S597-component-1 contacts the L1 domain of one IR protomer, while the component-2 from the same S597 interacts with the FnIII-1 domain of the adjacent protomer (Fig. 3c). In this way, one S597 molecule simultaneously engages the two IR protomers, thereby stabilizing the active conformation of IR. The linker connecting the site-1 and site-2 components of IR that consist of six residues was not resolved in the cryo-EM map,

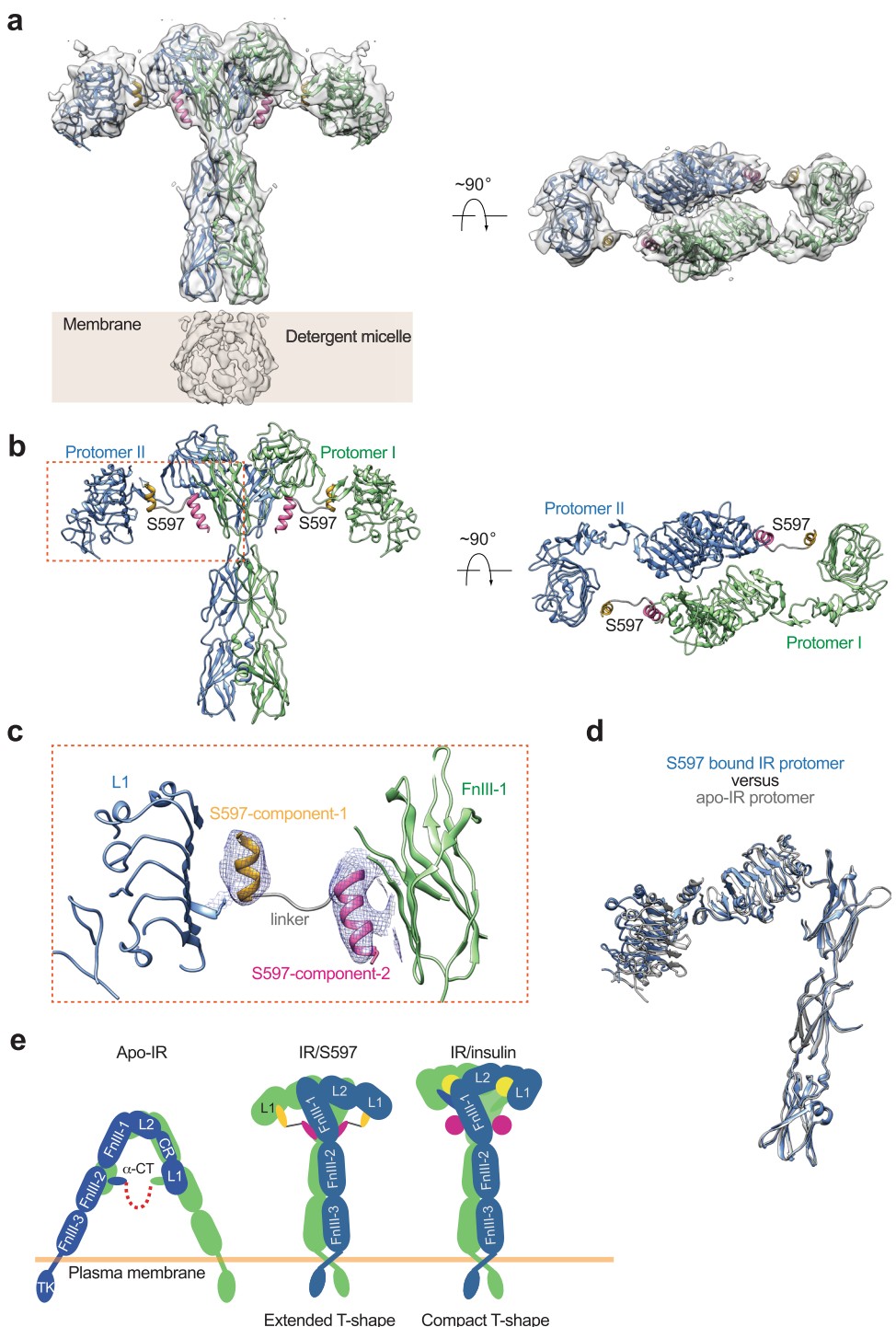

**Fig. 3 | Overall structure of the IR/S597 complex. a** 3D reconstruction of the IR dimer with two S597 peptides bound. **b** The ribbon representation of the IR dimer with two S597 peptides bound fitted into the cryo-EM map at 5.4 Å resolution. **c** Close-up view of the binding of S597 at the L1 domain of one protomer (blue) and FnIII-1 domain of another (green). The component-1 binding helix of S597 is shown in yellow, the component-2 binding helix in pink, and the linker region in gray.

**d** Superposition between S597-bound IR protomer (blue) and apo-IR protomer (gray). **e** Working model for S597-induced IR activation. The simultaneous binding of S597 to both the L1 domain of one protomer and the FnIII-1 domain of another would trigger the structural transition of IR directly from Λ-shaped apo-form to extended T-shaped IR dimer and stabilize the active form. A cartoon representation of the IR/insulin complex in a compact T-shape is shown for comparison.

reflecting its structural flexibility. The dynamic nature of S597 likely contributes to the structural flexibility of the entire IR/S597 complex, partially explaining why the cryo-EM map of the IR/S597 complex was limited in resolution. In addition to the S597-mediated protomer-protomer interaction, the L2 domain of one protomer also interacts weakly with the FnIII-1 domain of another, further maintaining the extended T-shaped active conformation of IR (Fig. 3a, b).

Our structural comparisons reveal that the conformation of one protomer in this extended T-shaped IR closely resembles that of the protomer in the Λ-shaped apo-IR dimer (Fig. 3d). This structural analysis strongly suggests that, after the apo-state of IR is disrupted by the binding of S597-component-1 to the L1 domain of IR, the two protomers undergo a rigid-body rotation using the L2-FnIII-1 interface as a hinge, leading to the extended T-shaped IR (Fig. 3e). This activation

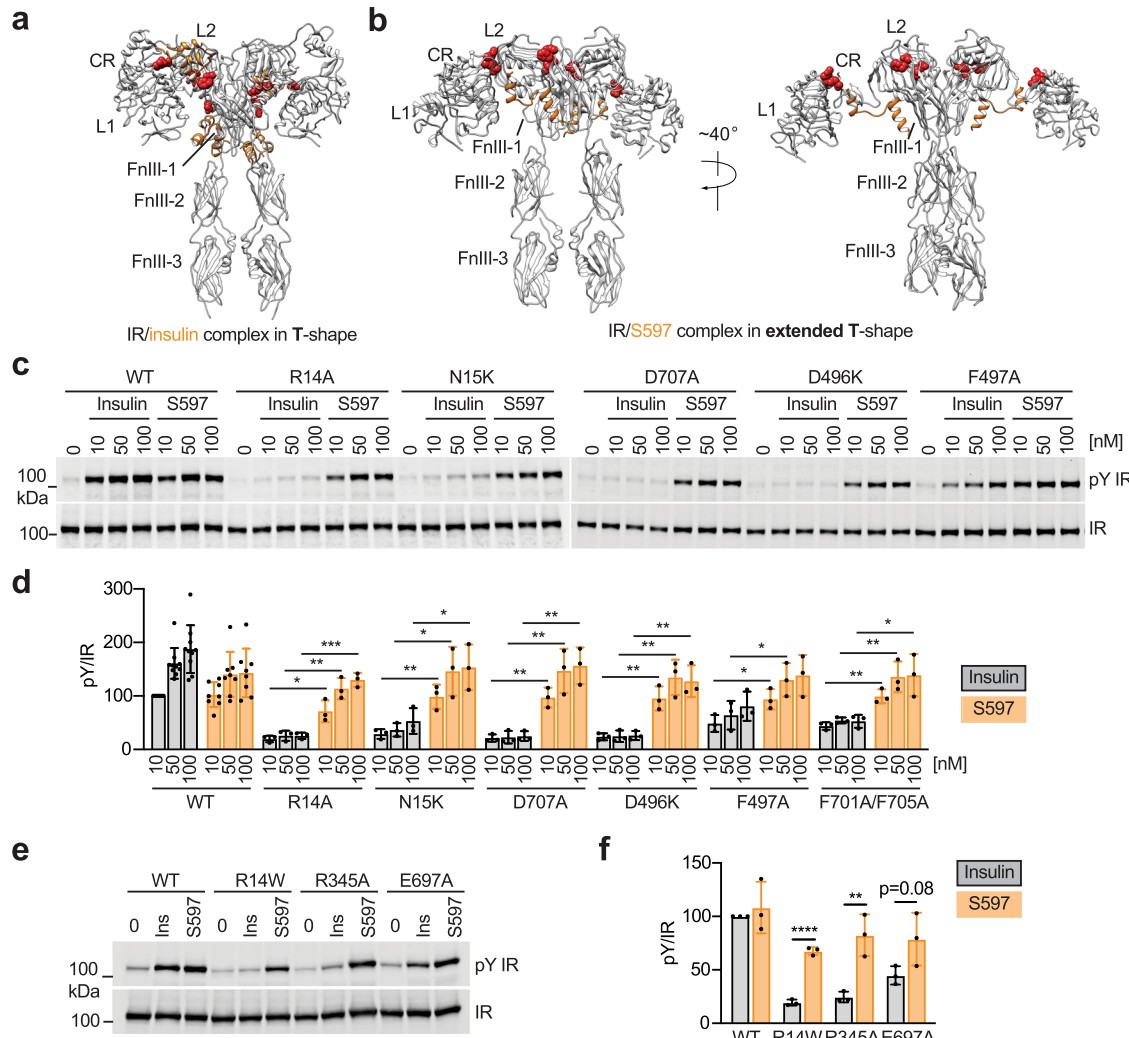

**Fig. 4 | S597 activates IR mutants that cause severe insulin resistance syndromes. a** Overall view of IR (gray)/insulin (orange) complex in a compact T-shape. The IR mutations we tested are indicated by red space-filling. **b** Overall view of IR (gray)/S597 (orange) complex in extended T-shape. The IR mutations we tested are indicated by red space-filling, except for E697, as we were not able to observe the α-CT in the IR/S597 complex. **c** Auto-phosphorylation of IR by the indicated concentrations of insulin or S597 for 10 min in 293FT cells expressing IR wild-type (WT) or indicated disease-causing mutants. **d** Quantification of the western blot data shown in **c**. Levels of pY IR were normalized to total IR levels and shown as intensities relative to that of IR WT in 10 nM insulin-treated cells. Mean ± SD. Each

experiment was repeated three times. Significance calculated using two-tailed Student's *t*-test; *$p < 0.05$, **$p < 0.01$, and ***$p < 0.001$. The exact *p* values are provided in the source data. **e** Auto-phosphorylation of IR by 10 nM insulin or S597 for 10 min in 293FT cells expressing IR WT or indicated disease-causing mutants. **f** Quantification of the western blot data shown in **e**. Levels of pY IR were normalized to total IR levels and shown as intensities relative to that of IR WT in insulin-treated cells. Mean ± SD. Each experiment was repeated three times. Significance calculated using two-tailed Student's *t*-test; **$p < 0.01$ and ****$p < 0.0001$. The exact *p* values are provided in the source data. Source data are provided as a Source Data file.

mechanism is distinct from the insulin-induced IR activation that involves both intra-protomer motion and inter-protomer rotation[20,24]. Furthermore, S597 stabilizes the active conformation of IR in a distinct manner from insulin. Specifically, in the T-shaped active IR/insulin complex, the site-1 insulin, which primarily binds to the L1/α-CT of one protomer, also contacts the top surface of the FnIII-1 domain of another[20,24]. In this way, the binding of two insulins to site-1 of IR brings the L1 domains on top of the FnIII-1 domains, thus triggering a compact T-shaped IR conformation. Two additional insulins bind to the side-surface of the FnIII-1 domains of IR, known as site-2. However, the site-2 insulins do not play major roles in maintaining the active conformation of IR, as they only contact one protomer. In contrast to the 4:1 (insulin: IR dimer) IR/insulin complex, only two S597s are bound to the IR dimer in the structure of the IR/S597 complex. Each S597 molecule simultaneously interacts with both the L1 domain and the side-surface of the FnIII-1 domain; such that the L1 domain of one protomer is placed on

the lateral side of the FnIII-1 domain of the adjacent protomer, leading to a more extended T-shaped active conformation.

## S597 activates disease-causing IR mutants

Certain mutations of IR cause inherited severe insulin resistance syndromes[8]. Because S597 binds and activates IR in a different manner from insulin, we explored whether S597 could activate those disease-causing IR mutants, which are predicted to be defective in insulin binding. We selected IR site-1 binding-deficient mutations that are found in the L1 (R14W[40] and N15K[41,42]), FnIII-1 loop (D496N[43]), and α-CT (D707A[20,44,45]) domains (Fig. 4a, b and Fig. S5), all of which are not localized at IR/S597 interfaces. The Asp496 in the IR was mutated to lysine as the charge reversion mutation was expected to disrupt insulin binding. The Arg14 in the IR was mutated to either alanine or tryptophan. In addition to these disease-causing IR mutants, we mutated several other key residues that are important for insulin binding but

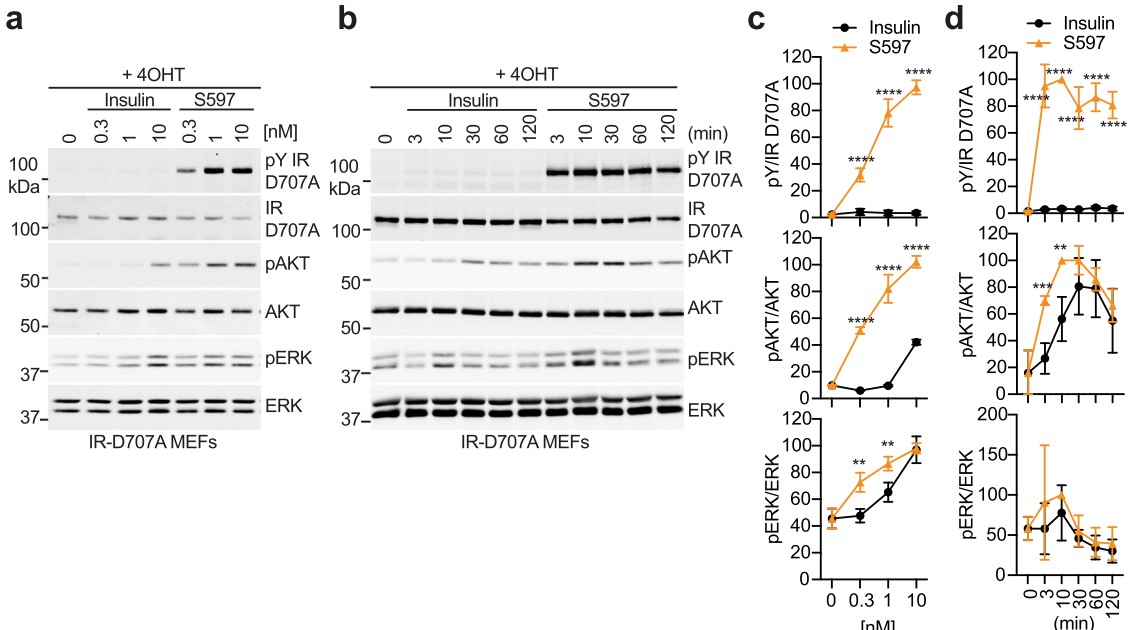

**Fig. 5 | S597 activates IR signaling in cells expressing disease-causing, insulin-binding-deficient IR mutants. a** IR signaling in IR-D707A MEFs treated with the indicated concentrations of insulin or S597 for 10 min. Cell lysates were blotted with the indicated antibodies. 4OHT, 4-Hydroxytamoxifen. **b** IR signaling in IR-D707A MEFs treated with 10 nM insulin or S597 for the indicated times. 4OHT, 4-Hydroxytamoxifen. **c** Quantification of the western blot data shown in **a**. Levels of protein phosphorylation were normalized to total protein levels and shown as intensities relative to that in 10 nM S597-treated cells. Mean ± SD. Each experiment was repeated four times. Significance calculated using two-tailed Student's *t*-test; between insulin and S597 in the indicated concentrations, **$p < 0.01$ and ****$p < 0.0001$. The exact *p* values are provided in the source data. **d** Quantification of the western blot data shown in **b**. Levels of protein phosphorylation were normalized to total protein levels and shown as intensities relative to that in 10 nM S597-treated cells for 10 min. Mean ± SD. Each experiment was repeated four times. Significance calculated using two-tailed Student's *t*-test; between insulin and S597 in the indicated time points, **$p < 0.01$, ***$p < 0.001$, and ****$p < 0.0001$. The exact *p* values are provided in the source data. Source data are provided as a Source Data file.

not S597 binding, including Phe497 in the loop of the FnIII-1 domain and Phe701/Phe705 of the α-CT motif (which contact the L1 domain) (Fig. 4c, d and Fig. S6c). IR R14W, R14A, N15K, D707A, D496K, and F701A/F705A showed expectedly reduced insulin-dependent IR activation, confirming their deficiency in insulin binding (Fig. 4c–f and Fig. S6c). IR F497A can be activated by insulin but with lesser efficiency than IR WT. Nevertheless, all the above IR mutants could be fully activated by S597 (Fig. 4c–f and Fig. S6c).

We have previously demonstrated that insulin-bound IR undergoes a large conformational change between the CR and L2 domains and between L2 and FnIII-1 domains[20]. These changes generate intra- and inter-domain contacts, thus stabilizing compact T-shaped IR. Due to the distinct activation mechanism of IR upon S597 binding, we hypothesized that S597 could activate IR mutants that disrupt the stabilization of the compact T-shaped IR upon insulin binding. To test this, we selected two residues, Arg345 (L2 domain) and Glu697 (N-terminal region of α-CT) in the IR, which form a salt bridge in the insulin-induced compact T-shaped IR. Mutation of IR R345A or E697A largely diminished IR activation by insulin, but those mutants could be fully activated by S597 (Fig. 4e, f). These results confirm our structural observation that S597 uniquely engages and activates IR and suggest that S597 activates disease-causing and insulin-binding-deficient mutants of IR, as well as mutants that disrupt the stability of the compact T-shaped IR.

In order to fully characterize the action between insulin and S597 in IR signaling, we created a cell-based model of severe insulin resistance. IR-D707A tagged with the C-terminal green fluorescence protein (GFP) or GFP alone was introduced into IR conditional knockout mouse embryonic fibroblasts (IR-D707A MEFs and GFP MEFs, respectively). We treated the cells with tamoxifen to remove endogenous murine IR and monitored auto-phosphorylation of IR (pY1152/1153; equivalent to pY1150/1151 of human IR, pY IR), AKT (pT308 and pAKT),

and ERK1/2 (pT202/Y204 and pERK) over a wide range of insulin concentrations (Fig. 5a, c and Fig. S8a, c) and at multiple time points (Fig. 5b, d and Fig. S8b, d). Tamoxifen treatment greatly reduced endogenous murine IR levels, but the residual IR could activate downstream signaling upon a high concentration of insulin stimulation (Fig. S8). The expression levels of IGF1R were not changed by tamoxifen treatment (Fig. S8). The C-terminal GFP tag of IR-D707A enabled the size shift of the IR-D707A on the blots to distinguish endogenous murine pY IR and pY IGF1R from ectopic human pY IR-D707A (Fig. 5a, b). Consistent with the results in 293FT cells expressing IR-D707A, insulin-triggered phosphorylation of IR was remarkably reduced in IR-D707A MEFs (Fig. 5). Consequently, the level of phosphorylation of AKT and ERK was greatly reduced in insulin-treated IR-D707A MEFs, indicative of defective IR signaling. A high concentration of insulin significantly increased pAKT and pERK in IR-D707A MEFs, potentially through the residual IR or cross-talk with IGF1R. In sharp contrast, S597 significantly increased the level of phosphorylation of IR, AKT, and ERK in IR-D707A MEFs over a wide range of concentrations (Fig. 5). These results suggest that S597 or other similar mimetics could activate the IR signaling in patients with insulin binding-deficient IR mutants.

### S597 selectively activates IR signaling within different tissues

The insulin-activated IR triggers two major signaling pathways: the PI3K-AKT pathway and the MAPK pathway that control metabolism and cell growth and proliferation[1]. Previous studies demonstrated that S597 selectively activates the PI3K-AKT pathway, leaving the MAPK pathway inactive in blood leukocytes, skeletal muscles, and adipose tissues of mice[32,35]. Further, in L6 myoblasts expressing human IR, the S597-associated IR remains at the cell surface for longer periods of time than insulin-associated IR, presumably via attenuated internalization[32]. To compare S597-induced IR signaling across

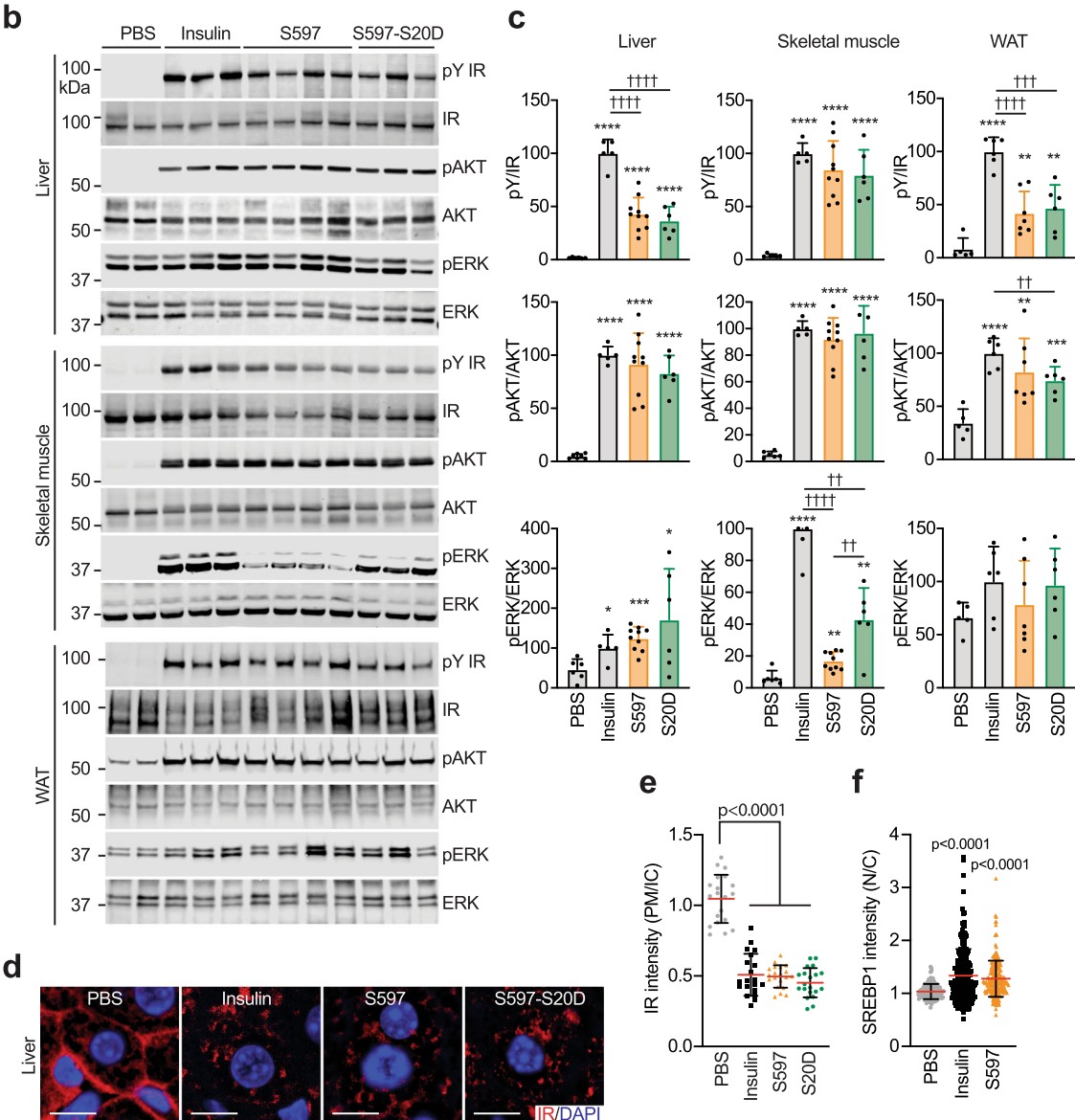

**Fig. 6 | S597 activates IR signaling in vivo. a** Sequences of S597 and S597-S20D. The residues in component-1 are marked in yellow, component-2 in pink, mutation in green, and linker in black. The disulfide bond was indicated as red. **b** Representative western blots of the liver, skeletal muscle, and white adipose tissue (WAT) lysates treated without or with insulin, S597 or S597-S20D. Each lane contains lysate from an individual mouse. **c** Quantification of the western blot data shown in **b**. Levels of protein phosphorylation were normalized to total protein levels and shown as intensities relative to that in insulin-treated conditions. Mean ± SD. PBS, $n = 6$ mice; insulin, $n = 5$; S597, $n = 10$; S597-S20D, $n = 6$ for liver and skeletal muscle. PBS, $n = 5$ mice; insulin, $n = 6$; S597, $n = 7$; S597-S20D, $n = 6$ for WAT. Significance calculated using two-tailed Student's $t$-test; *$p < 0.05$; **$p < 0.01$,

***$p < 0.001$, and ****$p < 0.0001$, with the PBS set as the control. ††$p < 0.01$ and ††††$p < 0.0001$. The exact $p$ values are provided in the source data. **d** Liver sections of mice injected without or with insulin or S597 analogs were stained with anti-IR (Red) antibodies and DAPI (blue). Scale bar, 10 μm. **e** Quantification of the ratios of the plasma membrane (PM) and intracellular compartment (IC) IR signals of the liver in **d**. Mean ± SD, Significance calculated using two-tailed Student's $t$-test. $N = 3$ mice each. **f** Quantification of insulin or S597-dependent nuclear translocation of SREBP1 in primary mouse hepatocytes. Representative images were shown in Fig. S9e. Mean ± SD. PBS, $n = 115$; Insulin, $n = 221$; S597, $n = 217$. Significance was calculated using a two-tailed Student's $t$-test; ****$p < 0.0001$, with the PBS set as the control. Source data are provided as a Source Data file.

different tissues, we examined the effect of S597 on IR signaling in the liver, epididymal white adipose tissue (WAT), and skeletal muscle of mice (Fig. 6b, c). S597 significantly increased the levels of pY IR and pAKT in skeletal muscle but was less potent in stimulating pERK (Fig. 6b, c). In the liver and WAT, S597 induced robust autophosphorylation of the IR, albeit with lower potency than insulin. In

contrast to the biased signaling observed in skeletal muscle, in the liver, S597 was able to increase both pAKT and pERK to similar levels as insulin (Fig. 6b, c). In the WAT, both insulin and S597 significantly increased pAKT levels, while neither insulin nor S597 had a statistically significant effect on pERK levels compared to vehicle-treated conditions. Moreover, like insulin, S597 promoted IR internalization in the

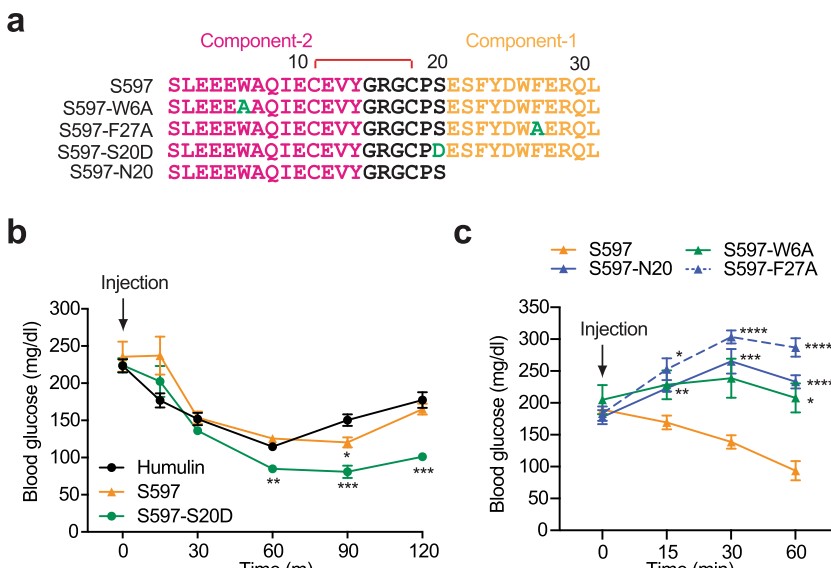

**Fig. 7 | S597 lowers blood glucose levels in mice. a** Sequences of S597, S597-W6A, S597-F27A, S597-S20D, and S597-N20. The residues in component-1 are marked in yellow, component-2 in pink, mutation in green, and linker in black. The disulfide bond was indicated as red. **b** Insulin tolerance test of 3-month-old male mice. Mice were injected intraperitoneally with Humulin (6 nmol/kg of body weight), S597 (9 nmol/kg), or S597-S20D (9 nmol/kg), and their blood glucose levels were measured at the indicated time points after injection. Mean ± SEM. Humulin, $n = 7$ mice; S597, $n = 6$; S597-S20D, $n = 7$. Significance was calculated using two-way ANOVA followed by Tukey's multiple comparisons test. *$p < 0.05$, **$p < 0.01$, and ***$p < 0.001$ with the Humulin set as the control. The exact $p$ values are provided in the source data. **c** Insulin tolerance test of 3-month-old male mice. Mice were injected intraperitoneally with S597 (9 nmol/kg), S597-W6A (9 nmol/kg), S597-F27A (9 nmol/kg), or S597-N20 (27 nmol/kg), and their blood glucose levels were measured at the indicated time points after injection. Mean ± SEM. S597, $n = 13$ mice; S597-W6A, $n = 5$; S597-F27A, $n = 5$; S597-N20, $n = 8$. Significance was calculated using two-way ANOVA followed by Tukey's multiple comparisons test. *$p < 0.05$, **$p < 0.01$, ***$p < 0.001$, and ****$p < 0.0001$ with the S597 set as the control. The exact $p$ values are provided in the source data. Source data are provided as a Source Data file.

liver of mice (Fig. 6d, e). We next isolated primary mouse hepatocytes and examined insulin- or S597-induced IR signaling over a wide range of insulin concentrations (Fig. S9a, b) and at multiple time points (Fig. S9c, d). As expected, in primary hepatocytes, S597-N20 (S597-component-2 alone) did not induce phosphorylation of IR, AKT, and ERK; while both S597 and insulin increased the levels of pY IR, pAKT, and pERK similarly, consistent with our in vivo data (Fig. S9a, b).

Insulin promotes nuclear translocation of sterol regulatory element-binding protein 1 (SREBP1) and stimulates hepatic lipogenesis[46]. Thus, we analyzed the cellular localization of SREBP1 in the absence or presence of S597 in primary hepatocytes (Fig. 6f and Fig. S9e). The nuclear translocation of SREBP1 was stimulated to the same level by insulin and S597, suggesting that S597 can activate SREBP1 function in the liver as well. These findings, together with a previous study in Zucker diabetic fatty rats[34] suggest that S597 promotes hepatic de novo lipid synthesis like insulin.

S597-component-1 and IR α-CT share high sequence similarity and bind the L1 domain of IR in a similar manner (Fig. 2d). In the structure of IR L1/α-CT, the salt bridge formed between Glu698 in the α-CT and Arg188 in the L1 domain maintains the α-helical conformation of the IR α-CT. Such interaction does not exist in the structure of IR L1/S597-component-1, as the residue corresponding to Glu698 of IR α-CT is serine in S597 (Ser20 S597) (Fig. 6a). Consequently, the S597-component-1 adopts a shorter α-helix with relative to IR α-CT (Fig. 2b, c). We hypothesized that substitution of the S597 serine at 20 to an aspartic or glutamic acid would form a salt bridge with the IR L1 domain, stabilizing the IR L1/S597-component-1 interaction and thereby modulating the agonist action of S597. To test this, we synthesized S597-S20D and examined its effects on IR activation (Fig. 6a). Like S597, S597-S20D increased the levels of pY IR and pAKT in the liver and skeletal muscle of mice (Fig. 6b, c), and promoted IR endocytosis in the liver (Fig. 6d, e). Interestingly, the pERK levels in both the liver and skeletal muscle of mice treated with S597-S20D were significantly

increased, in contrast to S597, which only increases the pERK levels in the liver but not in the skeletal muscle (Fig. 6b, c). Importantly, these results demonstrate that the signaling selectivity of S597 agonist action can be modulated by modest sequence changes.

Finally, to investigate the metabolic effects of S597 and each component of S597 in vivo, we performed insulin tolerance assays in mice (Fig. 7). Both S597 and S597-S20D lowered blood glucose in mice as effectively as insulin[34,35] (Fig. 7a, b), and S597-S20D displayed a longer lasting glucose lowering ability compared to S597. By contrast, individual administration of S597-N20 (S597-component-2 alone), S597-W6A (S597-component-2 mutant), and S597-F27A (S597-component-1 mutant) did not reduce glucose levels, but rather elevated them, likely due to the partial binding of S597 mutants preventing basal insulin from binding and activating the IR. These data confirm that both components-1 and -2 of S597 are required for IR activation (Fig. 7a, c). Taken together, these results further validate the function of S597 in IR signaling and show that S597 and a designed S597 analog (S597-S20D) can control metabolism as efficiently as insulin in mice.

## Discussion
### Mechanism of IR activation by S597
The binding of multiple insulins to the IR disrupts the auto-inhibited state of IR, inducing a large conformational change between the two protomers, and ultimately promoting a compact T-shaped active conformation, allowing the intracellular kinase domains to undergo efficient auto-phosphorylation[17,20,22,24]. Our structural and functional studies show that S597 can activate IR as efficiently as insulin but via an alternative mechanism. The binding of S597 to apo-IR first delocalizes the α-CT motif from the L1 domain by competing for the same binding surface on the L1 domain[36]. The delocalization of the α-CT motif then leads to the destabilization of the Λ-shaped auto-inhibited conformation of IR. Indeed, we did not observe the α-CT motif in our structure of the IR/S597 complex, which suggests that once the α-CT is released

from the L1 domain, it becomes disordered. Subsequently, the two protomers would undergo rigid-body rotation using the L2-FnIII-1 interface as the hinge. Finally, the active state of IR is stabilized by the concurrent binding of S597 to the L1 domain of one protomer and the FnIII-1 domain of another. In this binding mode, the L1 domain of one protomer is oriented laterally to the FnIII-1 domain of the adjacent protomer, resulting in an extended T-shaped symmetric conformation (Fig. 3). Notably, the IR/S597 complex is less rigid than IR/insulin complex, due to the flexible linker between the two components of S597. Nevertheless, the distance and orientation between two membrane-proximal domains are almost identical between IR/S597 and IR/insulin complexes, explaining why S597 can activate IR similarly to insulin (Fig. 4a, b).

### Selective IR signaling by S597 in various tissues

Our study demonstrates that S597 activates both the PI3K-AKT and MAPK pathways, and induces IR endocytosis in the liver, similarly to insulin. The liver is the major organ for insulin clearance mediated by IR endocytosis, and defects of this process can cause hyperinsulinemia[47–50]. The fact that S597 induces IR internalization suggests that S597 is cleared and elicits insulin-like signaling in the liver. Interestingly, in skeletal muscle, S597 selectively activates the PI3K-AKT pathways, while only weakly activating the MAPK pathway[32,35]. A recent study showed that S597 significantly weakens mitogenic signaling in monocytes and hematopoietic stem cells, and reduces local inflammation, thereby protecting atherosclerosis associated with metabolic syndrome in mice[35]. The mechanism by which insulin mimetics trigger selective signaling pathways (i.e., PI3K-AKT or MAPK pathways) in different cell types and tissues is unclear. Given the fact that the biased signaling is observed in skeletal muscle but not in the liver suggests that the relative expression of IR, IGF1R, and IR/IGF1R heterodimer contribute to the selectivity of IR signaling, as S597 specifically binds and activates IR, but not IGF1R[32]. Indeed, healthy hepatocytes or mature adipocytes do not express appreciable levels of IGF1R and predominantly express IR[51–53], whereas skeletal muscle cells express both IR and IGF1R[54]. Furthermore, although S597 did not change the IR auto-phosphorylation in cells overexpressing either IR-A or IR-B (Fig. S6a, b), we cannot rule out contributions from heterodimers formed between IR-A and IR-B (i.e., IR-A/IR-B), or even between IR isoforms and IGF1R (i.e., IR-A/IGF1R or IR-B/IGF1R) in vivo.

A limitation of our current in vivo study is that IR signaling in multiple tissues were measured at fixed time points and at a single dose. It is possible that the pharmacokinetics and signaling dynamics of S597 and S597 analogs may result in selective IR signaling in different tissues. Future studies will be required to determine whether structural differences of the IR induced by insulin or S597 contribute to this IR signaling selectivity in different tissues and metabolic conditions. Identifying S597 analogs, such as S597-S20D, that can activate both IR signaling pathways in skeletal muscle and then determining the structures bound to IR are needed to test this hypothesis.

### Comparison of S597 with monoclonal antibodies against IR

Monoclonal antibodies, such as 83-7 and 83-14, directly bind and activate native IR and several disease-causing IR mutants that express on the cell surface[45,55–57]. Although these antibodies alone can increase IR auto-phosphorylation by approximately 12% of the maximal native insulin response, they could effectively activate the PI3K-AKT pathways and lower blood glucose in mice, including mouse receptoropathy models. Similarly, monoclonal antibody XMetA acts allosterically and partially activates IR[58]. S597 is more potent than these monoclonal antibodies in activating native IR by about 80% of the maximum effect of insulin. We speculate that these IR monoclonal antibodies destabilize the auto-inhibitory state of IR by intramolecular cross-linking but cannot stabilize the active conformation as

effectively as insulin and S597 do, resulting in only partial activation of the IR.

A major problem with using monoclonal antibodies to activate the IR is the resultant downregulation of the IR[59], which can severely blunt the metabolic benefits. Previous studies have examined the chronic effects of S597 treatment in Zucker diabetic fatty rats and mice fed a diabetogenic diet, and have not demonstrated a downregulation of the functional IR when compared with chronic insulin treatment[34,35]. We also examined the chronic effects of S597 on the functional IR levels in two ways; (1) treatment with insulin or S597 at concentrations exceeding normal levels for 3 consecutive days (Fig. S10a–e) and (2) one hour after the treatment with physiological levels of insulin or S597[60] (Fig. S10f). The IR was not significantly downregulated in either condition. Given the high specificity of S597 for IR activation, a single chain peptide such as S597 may have distinct advantages over reagents that activate the IR through cross-linking.

### Insulin mimetics for insulin-desensitized IR

Together, our findings suggest that S597 and related insulin mimetics stabilize a structurally distinct active IR dimer and trigger insulin-like signaling. We also show that S597 can activate insulin binding-deficient IR mutants that cause severe insulin resistance syndromes as well as mutants that disrupt the stability of the insulin-induced active conformation of IR. These results raise the possibility that S597 restores IR signaling in patients with insulin-desensitized IR or even disease-causing IR mutants that disrupt insulin binding or stabilization of the compact T-shape of IR. It is worth to note that S597 is not expected to activate mutants of the IR that may cause misfolding or defects in trafficking or kinase activity. As S597 and S597 analogs provide a significant advantage over a previously described insulin mimetic that showed low receptor specificity[61,62], it will be intriguing to ask whether insulin in combination with S597 could lower insulin requirements in patients with diabetes[63]. Future studies are required to examine the long-term effects of S597 in vivo with insulin binding-deficient IR mutants and to validate whether such insulin mimetics can restore normal glucose and lipid metabolism in severe insulin resistance syndromes. Our structural models inform the design of S597-like peptides with stronger receptor binding affinity and higher structural stability. We expect such peptides to recapitulate insulin-like signaling and hold promise for the treatment of insulin-resistant conditions.

## Methods

### Mice

Animal work described in this manuscript has been approved and conducted under the oversight of the Columbia University Institutional Animal Care and Use Committee. Mice (C57BL/6 J, stock #000664, Jackson Laboratory) were fed a standard rodent chow (Lab diet, #5053). All animals were maintained in a specific antigen-free barrier facility (temperature, 68–79 °F; humidity, 30–70%) with 12 h light/dark cycles (6 a.m. on and 6 p.m. off). Two to three-month-old male mice were used in this study.

### Cell lines

**FreeStyle™ 293-F.** FreeStyle™ 293-F cells were obtained from Invitrogen (R79007). FreeStyle™ 293-F cells were cultured in FreeStyle™ 293 Expression Medium. FreeStyle™ 293-F cells were maintained in an orbital shaker in a 37 °C incubator with a humidified atmosphere of 5% $CO_2$.

**293FT.** 293FT cells were obtained from Invitrogen (R70007). 293FT cells were cultured in high-glucose Dulbecco's modified Eagle's medium (DMEM) supplemented with 10% (v/v) fetal bovine serum (FBS), 2 mM L-glutamine, and 1% penicillin/streptomycin. 293FT cells were maintained in monolayer culture at 37 °C and a 5% $CO_2$ incubator.

**Sf9 cells.** *Spdoptera frugiperda* (Sf9) cells were cultured in SF900 II SFM (Gibco) at 27 °C with orbital shaking at 120 rpm.

**Cell line validation.** An aliquot of each cell line was passaged for only 3–4 weeks, after which a fresh batch of cells was thawed and propagated. No mycoplasma contaminations were detected.

## Synthesis of insulin-mimetic peptides

**General reagents and methods.** All standard solvents and reagents were obtained from Sigma-Aldrich unless otherwise specified, Fmoc-protected amino acid were purchased from Vivitide, H-Rink-ChemMatrix resin (loading level: 0.47mmoml/g) from Gyros Technologies, trifluoroacetic acid (TFA) from EMD Millipore, piperidine from Alfa-Aesar, 1-hydroxy-6-chloro-benzotriazole (6-Cl-HOBt) from Creosalus-Advanced Chemtech, tri-isopropylsilane (TIS) from TCI. Analytical chromatography was performed on an Agilent 1100 model system equipped with an autosampler using Phenomenex columns as specified below. Preparative chromatography was conducted on a Waters Model 2525 binary pump system equipped with a Model 2487 absorbance detector. Mass spectral data was collected on a Waters Synapt G2 HDMS.

**Peptide synthesis.** The insulin-mimetic sequences were assembled on a 0.1 mmol scale by automated solid-phase methodology using an Applied Biosystems 431 A synthesizer on H-Rink-ChemMatrix support. The standard α-fluorenyl-methoxycarbonyl/t-butyl (Fmoc/tBu) based protecting group scheme was used: Arg(Pbf), Asp(OtBu), Cys(Trt), Glu(OtBu), Gln(Trt), Ser(tBu), Trp(Boc), Tyr(tBu). The automated cycles utilized 6-Cl-hydroxybenztriazole (6-Cl-HOBt)/ diisopropylcarbodiimide (DIC) for activation and 20% piperidine/DMF for deprotection. N-terminal acetylation was performed manually by treating the peptide resin with a DMF solution containing 5% acetic anhydride for 30 min. Resin cleavage and side-chain deprotection were carried in 15 ml of TFA containing 2% v/v of the following scavengers: water, TIS, thioanisole, and β-mercaptoethanol for 2.0 h. The crude peptide was recovered by the addition of excess cold diethyl ether, followed by several washes of the precipitate with diethyl ether and air drying.

**Oxidation/folding.** Oxidation to form the disulfide bridge involved dissolving the crude peptide in a 25 mM ammonium bicarbonate buffer (pH 7.7) at a concentration of 1.0 mg/ml and stirring the solution in an open beaker at room temperature. The progress of the oxidation was monitored by analytical HPLC chromatography using a Phenomenex Gemini® 5 μm C18 150 × 4.6 mm analytical column and a linear gradient of 10–40%B (0.1% TFA/ACN) over 30 min. The oxidation was complete after 24–36 h and the peptide was isolated by preparative HPLC chromatography using a Phenomenex Luna® 10 μm C8(2) 250 × 21.2 mm LC column and a 15–30%B gradient over 60 min. The homogeneity of the purified peptides was assessed by analytical HPLC and Mass Spectrometry to confirm product identity.

## Protein expression and purification

Protein expression and purification were performed following previous protocols[24]. Briefly, the cDNA fragment of the short isoform of full-length mouse IR (MmIR) with kinase-dead mutation (D1122N) and IR substrates binding and endocytosis defective mutation (Y962F) was cloned into the pEZT-BM expression vector. It is noteworthy to note that the ectodomains of human and mouse IR have a sequence homology of ~95.1% (Fig. S5). The HRV-3C protease recognition sequence and the protein Tsi3, which was used as a purification tag, were fused to the C-terminus. MmIR was expressed in FreeStyle™ 293-F (Invitrogen, #R79007) cells using the BacMam system following the standard protocol. Protein was expressed in FreeStyle™ 293-F cells by infecting the virus at a 1:10 (v:v) ratio. 6 h after infection, 8 mM sodium

butyrate was added to boost protein expression. The cells were cultured for 48–60 h at 30 °C.

The cells were resuspended in lysis buffer A containing 40 mM Tris-HCl pH 8.0, 400 mM NaCl, and a protease inhibitor cocktail (Roche). The membrane fraction was obtained by ultracentrifuge of the cell lysate for 1 h at 100,000×*g*. 1% Dodecylmaltoside (DDM) was added with stirring to extract the MmIR from the membrane fraction. The solubilized protein was obtained by ultracentrifuge again for 1 h at 100,000×*g*. The supernatant was added with 1 mM CaCl$_2$ and loaded to Tse3 protein conjugated Sepharose resin (GE Healthcare) by gravity flow. The resin was washed by wash buffer A (40 mM Tris-HCl, pH 8.0, 400 mM NaCl, 1 mM CaCl$_2$, 5% glycerol, and 0.08% DDM) and MmIR was eluted by HRV-3C protease cleavage. The protein was then run on Superose 6 increase 10/300 GL size-exclusion column (GE Healthcare) with buffer 20 mM Hepes-Na pH 7.4, 150 mM NaCl, 0.03% DDM. The dimer fraction was pooled and added with synthesized S597-component-2 or S597 at 1:4 (m:m). After incubation for half an hour, the protein was concentrated at 6 mg/ml for cryo-EM analyses. All the purification steps were performed at 4 °C.

## EM data acquisition

EM data acquisition, image processing, model building, and refinement were performed following previous protocols[24].

The cryo-EM grid was prepared by applying 3 μl of the protein samples (6 mg/ml) to glow-discharged Quantifoil R1.2/1.3 300-mesh gold holey carbon grids (Quantifoil, Micro Tools GmbH, Germany). Grids were blotted for 4.0 s under 100% humidity at 4 °C before being plunged into the liquid ethane using a Mark IV Vitrobot (FEI). Micrographs were acquired on a Titan Krios microscope (FEI) operated at 300 kV with a K3 direct electron detector (Gatan), using a slit width of 20 eV on a GIF-Quantum energy filter. SerialEM 3.8 was used for the data collection[64]. A calibrated magnification of 46,296 was used for imaging of IR/S597-component-2 sample, yielding a pixel size of 1.08 Å on a specimen. A calibrated magnification of 60,241 was used for imaging of IR/S597 sample, yielding a pixel size of 0.83 Å on a specimen. The defocus range was set from 1.6 to 2.6 μm. Each micrograph was dose-fractionated to 30 frames with a total dose of about 60 e⁻/Å². 

## Image processing

The cryo-EM refinement statists for both IR/S597-component-2 and IR/S597 datasets is summarized in Table 1. About 8388 movie frames of IR/S597-component-2 were motion-corrected and binned twofold, resulting in a pixel size of 1.08 Å, and dose-weighted using MotionCor2[65]. The CTF parameters were estimated using Gctf1.06[66]. RELION3 was used for the following processing[67]. Particles were first roughly picked by using the Laplacian-of-Gaussian blob method and then subjected to 2D classification. Class averages representing projections of IR/S597-component-2 in different orientations were used as templates for reference-based particle picking. Extracted particles were binned three times and subjected to 2D classification. Particles from the classes with fine structural features were selected for 3D classification using an initial model generated from a subset of the particles in RELION3. Particles from one of the resulting 3D classes showing good secondary structural features were selected and re-extracted into the original pixel size of 1.08 Å. Subsequently, we performed finer 3D classification with C2 symmetry imposed by using local search in combination with small angular sampling, resulting two additional good classes with improved density for the entire complex. The cryo-EM map after 3D refinement of the combined two classes was resolved at 3.6 Å resolution, but the stalk region of the complex appeared blurred, suggesting relative swinging motions between two halves of the complexes. To improve the resolution, we performed C2 symmetry expansion and focused refinement as described previously. The modified particle set was subjected to another round of 3D refinement with a soft mask around one-half of the complex,

**Table 1 | Cryo-EM data collection and refinement statistics**

|  | IR/S597-N20 EMD-27705 PDB: 8DTM | IR/S597 EMD-27704 PDB: 8DTL |
|---|---|---|
| **Data collection and processing** | | |
| Magnification | 46,296 | 60,241 |
| Voltage (kV) | 300 | 300 |
| Electron exposure (e$^-$/Å$^2$) | 60 | 60 |
| Defocus range (μm) | 1.6–2.6 | 1.6–2.6 |
| Pixel size (Å) | 1.08 | 0.83 |
| Symmetry imposed | C1 | C2 |
| Initial particle images (no.) | 2,789,169 | 1,282,550 |
| Final particle images (no.) | 135,709 | 152,565 |
| Map resolution (Å) | 3.5 | 5.4 |
| FSC threshold | 0.143 | 0.143 |
| **Refinement** | | |
| Initial model used (PDB code) | 4ZXB | 6PXV |
| Model composition | | |
| Nonhydrogen atoms | 6500 | 12,518 |
| Protein residues | 802 | 1544 |
| Ligands | | |
| R.m.s. deviations | | |
| Bond lengths (Å) | 0.009 | 0.003 |
| Bond angles (°) | 1.098 | 0.506 |
| **Validation** | | |
| MolProbity score | 1.99 | 1.82 |
| Clashscore | 10.23 | 9.6 |
| Poor rotamers (%) | 0 | 0 |
| Ramachandran plot | | |
| Favored (%) | 92.82 | 95.47 |
| Allowed (%) | 7.18 | 4.53 |
| Disallowed (%) | 0 | 0 |

leading to a markedly improved resolution for the entire half complex to 3.5 Å resolution.

About 4615 movie frames of IR/S597 were motion-corrected and binned twofold, resulting in a pixel size of 0.83 Å, and dose-weighted using MotionCor2. CTF correction was performed using Gctf. Particles were first roughly picked by using the Laplacian-of-Gaussian blob method and then subjected to 2D classification. Class averages representing projections of the IR/S597 in different orientations were used as templates for reference-based particle picking. A total of 1,282,550 particles were picked from 4615 micrographs. Particles were extracted and binned by three times (leading to 2.49 Å/pixel) and subjected to another round of 2D classification. Particles in good 2D classes were chosen (616,622 in total) for 3D classification using an initial model generated from a subset of the particles in RELION3. After initial 3D classification, one major class was identified, showing good secondary structural features. Particles from this good class were selected and re-extracted into the original pixel size of 0.83 Å. Subsequently, we performed finer 3D classification with C2 symmetry imposed by using local search in combination with small angular sampling, resulting one 5 good classes with similar conformation and 1 bad class with poor density throughout the entire IR/S597 complex. We combined five good classes, and the cryo-EM map after 3D refinement, CTF refinement, and particles polishing was resolved at 5.4 Å resolution.

## Model building and refinement
Model building of IR/S597-component-2 was initiated by rigid-body docking of the apo-state of apo-IR with minor adjustment. The model

of S597-component-2 was de novo built using Coot0.8.8[68]. Model building of IR/S597 was initiated by rigid-body docking of the structures of S597-component-1 bound L1, CR, L2, and S597-component-2 bound FnIII-1, FnIII-2, and FnIII-3 domains[20]. Both models were refined by using the real-space refinement module in the Phenix package (V1.17)[69]. Restraints on secondary structure, backbone Ramachandran angels, and residue side-chain rotamers were used during the refinement to improve the geometry of the model. MolProbity 4.5, as a part of the Phenix validation tools, was used for model validation (Table 1). Figures were generated in Chimera 1.15 and Pymol 2.3[70].

## Primary hepatocytes isolation
Primary hepatocytes were isolated from 2-month-old male mice with a standard two-step collagenase perfusion procedure as described earlier with some modifications[24,48,49]. Briefly, following anesthesia, the inferior vena cava was cannulated, and the liver was perfused with Liver Perfusion Medium (Thermo Scientific, Cat. #17701038) using a peristaltic pump (perfusion rate as 3 ml/min). After 1–2 s upon the appearance of white spots in the liver, we cut the portal vein with scissors to wash out blood, and then the liver was perfused with 30 ml of Liver Digest Medium (Thermo Scientific, Cat. #17703034). The dissected liver was gently washed with low-glucose DMEM and transferred to a sterile culture dish containing 15 ml Liver Digest Medium. The isolated liver cells were filtered through the 70 μm cell strainer into a 50 ml tube. After centrifuging at 50×$g$ for 5 min at 4 °C, cells were washed with cold low-glucose DMEM three times. Cells were resuspended with attached medium [Williams' Medium E supplemented with 5% (v/v) FBS, 10 nM insulin, 10 nM dexamethasone, and 1% penicillin/streptomycin] and plated on collagen (Sigma, #C3867)-coated dishes. After 2 h, the medium was changed to serum-free low-glucose DMEM. After 14–16 h, the cells were treated with insulin, S597, or S597-N20 to analyze IR signaling.

## Mouse embryonic fibroblast (MEFs) isolation and viral infection
$IR^{F/F}$ mice (Jackson #006955[71]) were crossed with CAG-CRE/ERT mice (Jackson #004682) to generate $IR^{F/F}$; CRE-ERT mice. The $IR^{F/F}$; CRE-ERT mice were backcrossed nine generations to C57BL/6 (Jackson #000664). The MEFs from $IR^{F/F}$; CRE-ERT mice were isolated as described earlier with some modifications[49]. E14.5 embryos were minced with sterile scalpels and incubated with 0.25% trypsin-EDTA for 10 min at 37 °C and a 5% CO$_2$ incubator. Fresh 0.25% trypsin-EDTA was added. After 10 min incubation, cells were put into fresh DMEM containing supplemented 10% (v/v) FBS, 2 mM L-glutamine, 55 μM β-mercaptoethanol, and 1% penicillin/streptomycin, and then centrifuged at 290×$g$ for 5 min. After the supernatant was removed, the cells were resuspended in the DMEM and incubated at 37 °C and 5% CO$_2$ incubator. The medium was changed the next day to remove debris and floating cells.

To generate IR-D707A MEFs or GFP MEFs, cDNAs encoding C-terminal GFP-tagged human IR-D707A or GFP alone was cloned into the pBabe-puro vector. pBabe-puro-IR-D707A-GFP or pBabe-puro-GFP, pCMV-gag/pol, and pCMV-VSV-G were transfected into 293FT cells using Lipofectamine™ 2000 (Invitrogen). Viral supernatants were collected at 2 days and 3 days after transfection, concentrated with a concentrator, and added to $IR^{F/F}$; CRE-ERT MEFs with 4 μg/ml of polybrene. Cells were selected with 2 μg/ml of puromycin 2 days after infection.

## Insulin receptor activation and signaling assay in cultured cells
The insulin receptor signaling assay were performed as described earlier with some modifications[20,24]. For the activation assay, the short isoform of human IR in pCS2-MYC was used as described previously[20,49]. Plasmid transfections into 293FT cells were performed with Lipofectamin™ 2000 (Invitrogen). One day later, the cells were serum starved for 14 hr and treated with the indicated concentrations

of insulin or S597 analogs. For IR-D707A MEFs, cells were incubated with 1 μM Tamoxifen (EMD Millipore, Cat. #508225) for 2 days, starved for 14 h, and treated with the insulin or S597 analogs.

The cells were incubated with the lysis buffer B [50 mM Hepes pH 7.4, 150 mM NaCl, 10% Glycerol, 1% Triton X-100, 1 mM EDTA, 100 mM sodium fluoride, 2 mM sodium orthovanadate, 20 mM sodium pyrophosphate, 0.5 mM dithiothreitol (DTT), 2 mM phenylmethylsulfonyl fluoride (PMSF)] supplemented with cOmplete Protease Inhibitor Cocktail (Roche), PhosSTOP (Sigma), and 25 U/ml turbo nuclease (Accelagen) on ice for 1 h. After centrifuging at 20,817×$g$ at 4 °C for 10 min, the concentrations of cell lysate were measured using Micro BCA Protein Assay Kit (Thermo Scientific). About 50–60 μg total proteins were analyzed by SDS-PAGE and quantitative Western blotting. The following antibodies were purchased from commercial sources: Anti-IR-pY1150/1151 (WB, 1:1000; 19H6; labeled as pY IR, Cat. #3024), anti-AKT (WB, 1:1000; 40D4, Cat. #2920), anti-pS473 AKT (WB, 1:1000; D9E, Cat. #4060), anti-ERK1/2 (WB, 1:1000; L34F12, Cat. #4696), anti-pERK1/2 (WB, 1:000; 197G2, Cat. #4377), anti-IRS1-pS616 (WB, 1:1000; C15H5, Cell Signaling, Cat. #3203); anti-IRS1 (WB, 1:1000; A301-158A, Bethyl Laboratory); anti-IR (WB, 1:500; CT-3, Santa Cruz; labeled as IR for primary hepatocytes and MEFs, Cat. #sc-57342); anti-IR (WB, 1:1000; EPR23566-103; labeled as IR for Fig. S10, ab278100, Abcam); anti-Myc (WB, 1:2000; 9E10, Cat. #11667149001, Roche; labeled as IR for 293FT cells). For quantitative western blots, anti-rabbit immunoglobulin G (IgG) (H + L) (WB, 1:5000; Dylight 800 conjugates, Cat. #5151) and anti-mouse IgG (H + L) (WB, 1:5000; Dylight 680 conjugates, Cat. #5470) (Cell Signaling) were used as secondary antibodies. The membranes were scanned with the Odyssey Infrared Imaging System (Li-COR, Lincoln, NE).

### Insulin signaling and insulin receptor endocytosis in vivo

Two to three months old male mice were fasted overnight. Following anesthesia, mice were injected with 6 nmol Humulin (Eli Lilly) or 9 nmol S597 analogs per mouse via inferior vena cava. Livers were removed 3 min after injection. White adipose tissue and skeletal muscles were removed at 5 and 7 min after injection, respectively. Tissues were mixed with the lysis buffer B supplemented with cOmplete Protease Inhibitor Cocktail (Roche), PhosSTOP (Sigma), and 25 U/ml turbo nuclease (Accelagen), homogenized with Fisherbrand™ Bead Mill homogenizer, and then incubated on ice for 1 h. After centrifuging at 20,817×$g$ at 4 °C for 30 min, the concentrations of cell lysate were measured using Micro BCA Protein Assay Kit (Thermo Scientific). The lysates were then analyzed by quantitative western blotting.

For IR endocytosis assays, the livers were fixed in 10% Neutral Buffered Formalin (NBF) and embedded in paraffin blocks. Sections were deparaffinized and subjected to immunohistochemistry as described in the immunohistochemistry section and previous study[48].

### Immunohistochemistry

The immunohistochemistry was performed as described earlier with some modifications[48,49]. Mouse tissues were fixed in 10% NBF and embedded in paraffin blocks by the Molecular Pathology Core at Columbia University. The deparaffinized sections were subjected to antigen retrieval with 10 mM sodium citrate (pH 6.0), incubated with 0.3% $H_2O_2$, blocked with 3% bovine serum albumin (BSA), and then incubated with anti-IR antibodies (CT-3, Millipore, Cat. # MABS65, 1:100) in 3% BSA in PBS with 0.1% Triton X-100 (0.1% PBST) at 4 °C overnight. The slides were washed and incubated with secondary antibodies (Alexa Fluor 568 donkey anti-mouse, Invitrogen, Cat. #A10037, 1:200). The washed slides were stained with 4′,6′-diamidino −2-phenylindole (DAPI). Ten images were randomly taken under x60 magnification with a Leica Thunder Imager (Leica Microsystems Inc.). Images of sections were acquired as a series of 0.2 μm stacks. The cell edges were defined with Image J as described earlier[48]. The whole cell

signal intensity (WC) and intracell cellular intensity (IC) were measured. The plasma membrane signal intensity (PM) was calculated by subtracting IC from WC. Identical exposure times and magnifications were used for all comparative analyses.

### Immunofluorescence assay for SREBP1 translocation

Primary hepatocytes were described in the primary hepatocyte isolation section. To analyze SREBP1 translocation, the isolated hepatocytes were resuspended with modified attached medium [Williams' Medium E supplemented with 10% (v/v) FBS, 100 pM insulin, 2 mM L-glutamine, 100 nM dexamethasone, 5.5 μg/mL transferrin, 6.7 ng/mL sodium selenite, and 1% penicillin/streptomycin] and plated on collagen-coated coverslips. After 14 h, cells were pre-treated with 10 μM MG-132 (MedChemExpress LLC, Cat. #HY-13259) for 2 h and then treated with 100 nM insulin (Sigma) or S597 for 30 min. Cells were fixed with 4% paraformaldehyde for 10 min. The fixed cells were incubated with PBS for 30 min and 3% BSA in PBS with 0.1% Triton X-100 (0.1% PBST) for 1 h, and then treated with anti-SREBP1 antibodies (20B12, Millipore, Cat. #MABS1987, 1:200) in 3% BSA in 0.1% PBST at 4 °C overnight. After being washed, cells were incubated with secondary antibodies (Alexa Fluor 488 goat anti-rabbit, Invitrogen, Cat. #A11008, 1:200) for 1 h. Cells were washed and mounted on microscope slides in ProLong Gold Antifade reagent with DAPI (Invitrogen). Images were acquired as a series of 0.2 μm stacks with a Leica Thunder Imager (Leica Microsystems Inc.). Raw images were deconvolved using LAS X Software (Leica Microsystems Inc.). The nuclear edges and the nuclear SREBP1 signal were defined with Image J. Identical exposure times and magnifications were used for all comparative analyses.

### Insulin tolerance test

Insulin tolerance tests were performed as described previously with some modification[48,49]. Briefly, 2–3 months old male mice were fasted for 2 h and their blood glucose levels (T = 0) were measured with tail bleeding using a glucometer (AlphaTRAK). After then, mice were injected intraperitoneally with Humulin (6 nmol/kg body weight), S597 (9 nmol/kg), S597-S20D (9 nmol/kg), S597-W6A (9 nmol/kg), S597-F27A (9 nmol/kg), or S597-N20 (27 nmol/kg), and their blood glucose levels were measured at the indicated time points after injection.

### Chronic S597 treatment and glucose tolerance test

For Fig. S10a–e, 2 months old male mice were injected intraperitoneally with a vehicle, insulin (50 U/kg body weight), or S597 (50 U/kg body weight) once daily for 3 days. For the oral glucose tolerance test, mice were fasted for 6 h and their blood glucose levels (T = 0) were measured with tail bleeding using a glucometer (AlphaTRAK). After then, glucose (2 g/kg body weight) was administered by oral gavage. Blood glucose levels were measured at the indicated time points after glucose administration. For Fig. S10f, 2 months old male mice were fasted for 14 h. Following anesthesia, mice were injected with vehicle (PBS), insulin (0.1 U/kg), S597 (0.15 U/kg), or S597-S20D (0.15 U/kg) via a jugular vein. One hour later, livers were collected.

### Statistical analysis

Prism 9 was used for the generation of graphs and for statistical analyses. Results are presented as Mean ± SD or Mean ± SEM. Two-tailed unpaired $t$-tests were used for pairwise significance analysis. Two-way ANOVA followed by Tukey's multiple comparisons test was used for insulin tolerance test analysis. Sample sizes were determined on the basis of the maximum number of mice. Power analysis for sample sizes were not performed. Randomization and blinding methods were not used, and data were analyzed after the completion of all data collection in each experiment.

## Data availability

All reagents generated in this study are available with a completed Materials Transfer Agreement. All cryo-EM maps and models reported in this work has been deposited into EMDB/PDB database under the entry ID: EMD-27704 (IR/S597), PDB 8DTL (IR/S597), EMD-27705 (IR/S597-N20), and PDB 8DTM (IR/S597-N20). PDB structures used in this study are as follows: 4ZXB and 6PXV. Source data are provided with this paper.

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

## Acknowledgements

Cryo-EM data were collected at the University of Texas Southwestern Medical Center (UTSW) Cryo-Electron Microscopy Facility, funded in part by the Cancer Prevention and Research Institute of Texas (CPRIT) Core Facility Support Award PR170644. We thank Dr. Stoddard for facility access and Drs. Rebecca Haeusler and Julie Canman for helpful discussions. We are grateful to the Molecular Pathology Core for assistance with tissue processing and sectioning. This work is supported in part by grants from the National Institutes of Health (R35GM142937 and UL1TR001873 to E.C., R01GM136976 to X.-c.B., AG061829 to M.H.B.S, and P30DK063608 to D.A.), the Welch Foundation (I-1944 to X.-c.B.), CPRIT (RP160082 to X.-c.B.), the MCDB Neurodegenerative Disease Fund (M.H.B.S), the T. Curtius Peptide Facility (M.H.B.S.) and the Alice Bohmfalk Charitable (to E.C.). X.-c.B. is Virginia Murchison Linthicum Scholar in Medical Research at UTSW. We thank Lauge Schäffer for insight and foundational contributions to the discovery of insulin mimetic peptides.

## Author contributions

X.-c.B., M.H.B.S., and E.C. designed and supervised research; J.P., D.A., and E.C. performed and analyzed animal experiments, J.L. prepared the sample for structural determination; J.P.M. and K.A.B. synthesized and purified peptides; J.W., C.H., and E.C. performed and analyzed cellular assays; X.-c.B. performed cryo-EM structure determination and model building. All authors participated in the paper preparation.; X.-c.B., M.H.B.S., and E.C. wrote the paper with input from other authors.

## Competing interests

The authors declare no competing interests.
