## [Peer Review File · Nature Communications]

Activation of the insulin receptor by an insulin mimetic peptideReviewer #1 (Remarks to the Author):

In this manuscript, Park and colleagues present a complete cryo-EM structural study of the activation of the insulin receptor by the insulin-non-related agonist peptide S597; how it binds to the receptor differently from insulin and how it stabilises an activated conformation of the receptor different from the one triggered by insulin. These findings are completed and validated by a series of in vivo activity experiments including clear evidences that this peptide can activate disease-related mutated insulin receptor unable to be activate by insulin itself.

This study is very clearly written, the figures supporting very well the "story". The results presented here are of high interest for the receptor tyrosine kinase community and push further the understanding of the complex activation mechanisms of the insulin receptor by non-insulin peptides, opening the way to future treatments for disease affecting this pathway.

I strongly support the acceptance of the manuscript in Nature Communications.

Reviewer #2 (Remarks to the Author):

The manuscript 'Activation mechanism of non-insulin peptide agonists at the insulin receptor' describes the cryo-EM structure of the insulin mimetic S597 bound structure of the exon 11 negative A-isoform of the insulin receptor (IR). The manuscript also examines the potential of S597 to activate mutant IR that lack the ability to bind insulin as a proof-of-concept to the use of S597 as a treatment for rare syndromes of severe insulin resistance. Severe insulin resistance syndromes lack effective therapeutics, thus IR receptor agonists that overcome mutations that underlie these syndromes could be potentially life saving. The authors have previously described an insulin bound structure of the IR and cryo-EM work in the current manuscript appears robust, with the S597-bound IR adopting a relatively flexible extended-T shape distinct to the insulin-bound IR. The cellular and in vivo studies are not quite as robust.

Specific comments are listed below:

Line 181: "This structural feature suggests that IR bound with the full-length S597 is trapped in the active state". What do the authors mean by "trapped"? 'Trapped' would imply that once bound, S597 does not dissociate, is this the case? Please clarify.

I appreciate that aCT and the insert domain is difficult to discern in structures, but given that S597 essentially competes for the aCT binding site, what happens to aCT in the S597 bound structure?

Please consider an addition to Figure 3e to include a comparison between insulin bound and S597 bound structures. This would help with the visualisation of the information provided by the text lines 199 - 213.

A cellular model of insulin receptoropathy was made by the expression of GFP-tagged IR D707A in a tamoxifen inducible knockout of endogenous mouse IR MEF cell line. However, no data is provided outlining the characterisation of this cell line. The antibodies stated in the methods (Line 910) for the detection of IR in these cell lines is CT-3, which would detect any remaining endogenous IR as well as the introduced D707A. Furthermore, there is considerable signalling downstream of insulin-stimulated D707A that occurs in a time and dose-dependent fashion (Figure 4 e-h). D707A does not bind insulin, this would suggest that there is either low-level endogenous WT IR expressed in these cells, or significant IGF1R expression. The authors describe the signalling as 'marginal' (line 242) however densitometry data (Figure g and h) demonstrate that pERK in response to insulin is equal to that of S597 at 10nM and throughout the time-course. Phospho-AKT in response to 10nM insulin is 40% that of S597 and equal to that of S597 at 30,60,120min time points. The authors speculate this

is through cross-talk with the IGF1R, however if appropriate controls had been performed then this could have been determined empirically.

- Could the authors please revise the use of the term 'marginal'.
- Could the authors please include data on the characterisation of this cell line, including the key control of IR and IGF1R expression and downstream signalling in tamoxifen-treated MEFs that do not express D707A.

Line 257-258. "Consistent with previous results, S597 significantly increased the levels of pY IR and pAKT in skeletal muscle and WAT but was less potent in stimulating pERK" I agree with this comment for skeletal muscle. But this does not appear to be the case for WAT. The Western blots in Extended data figure 6a, demonstrate considerable activation of ERK, if the data normalised for total ERK (the last two lanes of the blot appear to have lower detection of total ERK). The downstream signalling response to insulin and S597 in WAT should be presented for the whole cohort (like Figure 5c). Not only is the data different to previous published studies (ref 30), which is at odds with the statement on line 256 "Consistent with previous results,..." it potentially gives important insight into signalling responses of tissues with different IR and IGF1R expression profiles. Liver and mature adipose tissue predominantly express the IR-B isoform, with little to no expression of the IGF1R. Skeletal muscle expresses both IR-A and IR-B, along with the IGF1R. These tissues demonstrate differential downstream activation in response to S597, where ERK is phosphorylated in tissues that express IR-B without IGF1R (liver and WAT), whereas the tissue that expresses the full plethora of IR-A, IR-B, IGF1R and hybrid receptors (skeletal muscle) does not have an appreciable pERK response to S597. This is an important observation and should be discussed in the manuscript.

- Could the authors please:
 - include the full signalling dataset obtained from WET
 - review the statement "Consistent with previous results" (line 256)
 - discuss the differential signalling responses observed in tissues in context of IR-A, IR-B, IGF1R, and hybrid receptor expression.

Similar to the above, the S597 bound structure is the short exon 11 negative IR-A isoform. The authors note there is flexibility in the structure. The IR-B isoform contains a longer insert domain owing to the amino acids encoded by exon 11. It would be intriguing to understand any structural differences between the two IR isoforms upon either insulin or S597 binding. Especially in light of the differential downstream signalling activation in different tissues that are associated with different expression of IR isoforms, IGF1R and consequently IR/IGF1R hybrid receptors.

The insulin tolerance test data in Figure 5 g and h only extends to 60 minutes. Longer time courses such as 120 minutes are conventional and usually demonstrate a plateauing or rise of blood glucose at later time points. Were longer time-points collected? If not, why not? It would be important to understand whether blood glucose levels continue to fall at later time-points in S597 and S597-S20D treated mice.

- Could the authors please address the following:
 - Statistics are only provided for S597 analogue treated mice where blood glucose was elevated in response to administration. Presumably elevated blood glucose in S597 analogue treated mice (5g) is due to the partial binding of the binding site mutant analogues preventing endogenous insulin from binding and activating the IR, however this is not discussed in the manuscript.
 - It is not clear as to what was used as the comparator for statistical significance. Please clarify this.
 - The figure suggests that insulin, S597 and S597-S20D administration did not statistically decrease blood glucose levels in these mice. Was this the case?
 - Also, the strain of mice used for the in vivo studies is not stated in the manuscript. These details should be provided.

The data demonstrate potential utility of S597 to activate insulin-binding deficient mutants. In light of the distinct IR structural changes that S597 binding causes, it would

be important to examine the effects of S597 on IR mutants that perturb IR function in different ways, ie. mutations that occur away from the insulin binding site and perturb movement of hinge regions, and alter internalisation/recycling kinetics. This would demonstrate wider utility beyond mutations that affect insulin binding sites.

Minor Comments:

Inaccuracy in reporting of INSR mutants and cited literature. There are different substitutions for Arg14 described in ref 35 (R14W) and ref 20 (R14E). N15K is not described in ref 35. D496K is D496N in ref 36. Could this please be corrected.

Many of the figure legends throughout the manuscript do not state as to what the comparator was for the determination of statistical significance. Could this please be corrected.

Figure 2g-j, concentrations and timescale of treatment stimulations of insulin and S597 are not indicated in the figure, the figure legend, or associated text. Please amend.

Figure 2h, please update figure legend to indicate that KE/LA refers to K484E/L552A

Scale bar is not defined in extended data figure 6.

Reviewer #3 (Remarks to the Author):

This study addresses S597, a single chain insulin receptor agonist developed around 20 years ago and studied intermittently since then. S597 has interesting signalling properties in cells, having been reported previously to favour metabolic over mitogenic signalling, and has also been shown to exert beneficial acute metabolic effects in mice.

This study moves understanding forward first by reporting use of cryoEM to solve a series of structures including the mouse insulin receptor and all or parts of S597, and second by providing evidence that the peptide can be used to activate naturally occurring mutated receptors, with potential implications for precision therapy for highly selected forms of insulin resistance caused by INSR mutations.

I found the paper generally well written and interesting. In places I thought the context in which the work is presented could be improved to make the study more accessible to general readers. I offer specific comments below. I am not a structural biologist, so I focus my comments in particular on signalling studies and studies of receptor mutations:

- Can the authors comment and justify why the short isoform of the insulin receptor was used for their studies? This is generally regarded as the less "metabolic" of the INSR isoforms, based in part on tissue expression patterns. Potential importance of INSR/IGF1R hybrids is involved in discussion of biased agonism, yet any differences in IR-A vs IR-B action could be just as relevant. Is there any structural reason to suppose differential effects of S597 on the two isoforms?

- On a similar note, it is the mouse receptor that is used for structure determination, but cellular studies are undertaken using human INSR. This is clear from the methods but is not very explicit in results. Given high homology between species it is almost certain that findings between mouse/human are generalisable, but it may be worth commenting on conservation of key residues at the S597 binding sites.

- The authors might consider adding a further cartoon or similar to Fig 3 to illustrate how the S597-bound structure is similar and different to the insulin-bound structure.

●The study focuses on an informative group of INSR mutations that are said to have been found to be disease-causing in humans. However this is not completely clear from the references given. The human genetic reference 35 given for R14A seems to describe R14W (R41W) only on quick reading ([doi:10.1007/s00592-013-0490-x](https://doi.org/10.1007/s00592-013-0490-x)) while I could not see N15K mentioned at all in that article. Descriptions of identification of D496K and D707A are clear. As new mutations continue to be discovered in recessive INSR defects I don't think it is critical that all mutations studied have been previously described in humans – this study is more about proof of concept. However could the authors please make sure their citation is accurate.

●Based on the structures they present, can the authors offer more explicit comment on what types of missense mutations would be amenable to activation by S597, and which wouldn't, in all likelihood? Obviously only those which are expressed at the cell surface rather than being misfolded and degraded will be accessible.

●The paper perpetuates the long standing notion of a 'metabolic' pathway downstream from the INSR mediated by PI3K/AKT, and a "Mitogenic" pathway mediated by MAPK. However growth is really a metabolic phenomenon, the PI3K/AKT pathway is close to the most mutationally activated signalling pathway in cancer, and in any case there is clear crosstalk between RAS and PI3K, including direct binding. Is it time to downplay this simplistic dichotomisation?

●The study shows that in some contexts S597 activates both PI3K/AKT and RAS/MEK/ERK signalling, but in others in vivo activation only of PI3K/AKT occurs. Very limited discussion of this focuses on a possible role of INSR/IGF1R heterodimers, which differ among tissues based on expression patterns of INSR and IGF1R. No data are shown to address this possibility, and as mentioned above IR-A/IR-B ratios could also be involved. Many other explanations are also possible, for example relating to the pharmacokinetics and signalling dynamics of S597 – what concentrations reach different interstitial spaces, and what are the effects on recycling dynamics etc on chronic exposure? This is speculation only, but at present I think the discussion is too simplistic and may be made more nuanced

●A previously reported approach to activating wild type and mutant insulin receptors, dating from the early 1990s, relies on bivalent ligands such as intact antibodies, whose stimulatory activity relies on bivalency and receptor cross linking. This can activate receptors in cells and induce some glucose lowering in animal models, including models of receptoropathy (e.g. PMID 32816962). One problem with that approach is that receptor downregulation results, severely blunting metabolic benefits. In this paper, brief acknowledgement of this longstanding, parallel line of work seeking to activate mutant antibodies may benefit readers. Do the authors have any evidence that chronic treatment with S597 is more or less likely than chronic insulin treatment to downregulate cell surface receptor? A monovalent single chain peptide such as S597 may well have distinct advantages over crude reagents that activate by cross linking, and it would be interesting to note this.

●It is assumed that there may be a specific structural basis for biased agonism, but similar biased agonism is seen for the bivalent antibodies mentioned above. This makes me wonder if there is a more generalisable, kinetic explanation for biased agonism induced by different types of ligand, to re-iterate one of the points made above?

●In their summing up at the end of the discussion, the authors comment briefly that "As S597 or S597 analogs provide a significant advantage over a previously described insulin mimetic that showed low receptor specificity, it will be intriguing to ask whether insulin in combination with S597 could lower insulin requirements in patients with diabetes, decreasing the risk of hypoglycemia, weight gain, and oncogenesis associated with insulin therapy alone", citing a 2012 review from when the debate about oncogenicity of currently used insulin was live. This simple and very speculative statement masks a great deal of clinical and epidemiological complexity and looks very

alarming to the uninitiated regarding chronic insulin use. The authors do not elaborate on how S597 might have these beneficial effects. I suggest toning it down a little.

We thank reviewers for positive and constructive comments. In the revised manuscript, we have addressed all the reviewers' concerns by performing more experiments and substantially rewriting the manuscript.

Our point-by-point responses are listed below. For ease of reading, we have colored our responses in blue.

REVIEWER COMMENTS

Reviewer #1 (Remarks to the Author):

In this manuscript, Park and colleagues present a complete cryo-EM structural study of the activation of the insulin receptor by the insulin-non-related agonist peptide S597; how it binds to the receptor differently from insulin and how it stabilises an activated conformation of the receptor different from the one triggered by insulin. These findings are completed and validated by a series of in vivo activity experiments including clear evidences that this peptide can activate disease related mutated insulin receptor unable to be activate by insulin itself. This study is very clearly written, the figures supporting very well the "story". The results presented here are of high interest for the receptor tyrosine kinase community and push further the understanding of the complex activation mechanisms of the insulin receptor by non-insulin peptides, opening the way to future treatments for disease affecting this pathway.

I strongly support the acceptance of the manuscript in Nature Communications.

Response: This is very clear and succinct summary of our work. We thank the reviewer for the positive assessment of our manuscript.

Reviewer #2 (Remarks to the Author):

The manuscript 'Activation mechanism of non-insulin peptide agonists at the insulin receptor' describes the cryo-EM structure of the insulin mimetic S597 bound structure of the exon 11 negative A-isoform of the insulin receptor (IR). The manuscript also examines the potential of S597 to activate mutant IR that lack the ability to bind insulin as a proof-of-concept to the use of S597 as a treatment for rare syndromes of severe insulin resistance. Severe insulin resistance syndromes lack effective therapeutics, thus IR receptor agonists that overcome mutations that underlie these syndromes could be potentially life saving. The authors have previously described an insulin bound structure of the IR and cryo-EM work in the current manuscript appears robust, with the S597-bound IR adopting a relatively flexible extended-T shape distinct to the insulin bound IR. The cellular and in vivo studies are not quite as robust.

Response: We thank the reviewer for the positive assessment of our manuscript, and we greatly appreciate the constructive comments which we have addressed below.

Specific comments are listed below:

Line 181: "This structural feature suggests that IR bound with the full-length S597 is trapped in the active state". What do the authors mean by "trapped"? 'Trapped' would imply that once bound, S597 does not dissociate, is this the case? Please clarify.

Response: We used “trapped” to describe that the S597 stabilizes the active conformation of IR. We revised this description to the following “This structural feature suggests that IR bound with the full-length S597 represents the active ligand-receptor complex”.

I appreciate that aCT and the insert domain is difficult to discern in structures, but given that S597 essentially competes for the aCT binding site, what happens to aCT in the S597 bound structure?

Response: As the reviewer pointed out, we did not observe the insert domain and aCT in the IR bound with the full-length S597. We expect that the aCT is disordered after release from the L1 domain, and this delocalization of the aCT would destabilize the auto-inhibitory conformation of IR. This point has been further discussed in the main text.

Please consider an addition to Figure 3e to include a comparison between insulin bound and S597 bound structures. This would help with the visualisation of the information provided by the text lines 199 – 213.

Response: We thank the reviewer for making this good suggestion. As suggested, we have added insulin-bound IR dimer in Figure 3e.

A cellular model of insulin receptoropathy was made by the expression of GFP-tagged IR D707A in a tamoxifen inducible knockout of endogenous mouse IR MEF cell line. However, no data is provided outlining the characterisation of this cell line. The antibodies stated in the methods (Line 910) for the detection of IR in these cell lines is CT-3, which would detect any remaining endogenous IR as well as the introduced D707A. Furthermore, there is considerable signalling downstream of insulin-stimulated D707A that occurs in a time and dose-dependent fashion (Figure 4 e-h). D707A does not bind insulin, this would suggest that there is either low-level endogenous WT IR expressed in these cells, or significant IGF1R expression. The authors describe the signalling as ‘marginal’ (line 242) however densitometry data (Figure g and h) demonstrate that pERK in response to insulin is equal to that of S597 at 10nM and throughout the time-course. Phospho-AKT in response to 10nM insulin is 40% that of S597 and equal to that of S597 at 30,60,120min time points. The authors speculate this is through cross-talk with the IGF1R, however if appropriate controls had been performed then this could have been determined empirically.

•Could the authors please revise the use of the term ‘marginal’?

Response: Point accepted. We have made the correction.

• Could the authors please include data on the characterisation of this cell line, including the key control of IR and IGF1R expression and downstream signalling in tamoxifen-treated MEFs that do not express D707A?

Response: We thank the reviewer for making this great suggestion. We have examined the levels of IR and IGF1R and downstream signaling in the tamoxifen-treated MEFs expressing GFP only. Tamoxifen treatment greatly reduced endogenous murine IR levels, but the residual IR could activate downstream signaling upon high concentration of insulin stimulation (**Fig. S8**). The expression levels of IGF1R were not changed by tamoxifen treatment (**Fig. S8**). It is worth to note that the C-terminal GFP tag of IR D707A enabled the size shift of the IR D707A on the blots to distinguish endogenous murine pY IR and pY IGF1R from ectopic human pY IR D707A (**Fig. 5a,b**). Insulin-triggered phosphorylation of IR was remarkably reduced in IR-D707A MEFs (**Fig. 5**). Consequently, the level of phosphorylation of AKT and ERK was greatly reduced in insulin-

treated IR-D707A MEFs, indicative of defective IR signaling. High concentration of insulin significantly increased pAKT and pERK in IR-D707A MEFs potentially through the residual IR or cross-talk with IGF1R. In sharp contrast, S597 significantly increased the level of phosphorylation of IR, AKT, and ERK in IR-D707A MEFs over a wide range of concentrations (**Fig. 5**). These results suggest that S597 or other similar mimetics could activate the IR signaling in patients with insulin binding-deficient IR mutants.

Line 257-258. “Consistent with previous results, S597 significantly increased the levels of pY IR and pAKT in skeletal muscle and WAT but was less potent in stimulating pERK” I agree with this comment for skeletal muscle. But this does not appear to be the case for WAT. The Western blots in Extended data figure 6a, demonstrate considerable activation of ERK, if the data normalized for total ERK (the last two lanes of the blot appear to have lower detection of total ERK). The downstream signalling response to insulin and S597 in WAT should be presented for the whole cohort (like Figure 5c). Not only is the data different to previous published studies (ref 30), which is at odds with the statement on line 256 “Consistent with previous results,...” it potentially gives important insight into signalling responses of tissues with different IR and IGF1R expression profiles. Liver and mature adipose tissue predominantly express the IR-B isoform, with little to no expression of the IGF1R. Skeletal muscle expresses both IR-A and IR-B, along with the IGF1R. These tissues demonstrate differential downstream activation in response to S597, where ERK is phosphorylated in tissues that express IR-B without IGF1R (liver and WAT), whereas the tissue that expresses the full plethora of IR-A, IR-B, IGF1R and hybrid receptors (skeletal muscle) does not have an appreciable pERK response to S597. This is an important observation and should be discussed in the manuscript.

• Could the authors please:

- include the full signalling dataset obtained from WET

Response: We thank the reviewer for pointing out this important point. We have included the full signaling dataset obtained from WAT, and the downstream signaling response to insulin and S597 has been presented for the whole cohort, similar to liver and skeletal muscle (**Fig. 6b,c**). In the liver and WAT, S597 induced robust auto-phosphorylation of the IR, albeit with lower potency than insulin. In contrast to the biased signaling observed in skeletal muscle, in liver, S597 was able to increase both pAKT and pERK to similar levels as insulin. In the WAT, both insulin and S597 significantly increased pAKT levels, while neither insulin nor S597 had a statistically significant effect on pERK levels, compared to vehicle treated conditions.

- review the statement “Consistent with previous results” (line 256)

Response: Point accepted. We have made the correction.

- discuss the differential signalling responses observed in tissues in context of IR-A, IR-B, IGF1R, and hybrid receptor expression.

Response: This revised manuscript discusses biased signaling activation by S597 stimulation in different tissues in the context of IR isoforms, IGF1R, and IR/IGF1R hybrid expression. We also tested whether S597 differentially activates the IR-A and IR-B. We expressed IR-A and IR-B in 293FT cells and examined the IR auto-phosphorylation over a wide range of concentrations of insulin and S597 (**Fig. S6a,b**). Our data demonstrated that S597 increases the auto-phosphorylation of IR-A and IR-B to a similar level as insulin. The S597-activated IR-A and IR-B increased pAKT to a similar level as insulin-activated IR-A and IR-B. Interestingly, both S597-activated IR-A and IR-B were less potent than insulin in increasing pERK levels. These data suggest that S597 activates both IR-A and IR-B in a similar manner. Nevertheless, we cannot rule out contributions from heterodimers formed between IR-A and IR-B, or between IR isoforms and

IGF1R *in vivo*. Further studies are required to examine the effects of S597 on IR signaling and physiological functions in animal models expressing only IR-A or IR-B. We have included new data and expanded the discussion.

Similar to the above, the S597 bound structure is the short exon 11 negative IR-A isoform. The authors note there is flexibility in the structure. The IR-B isoform contains a longer insert domain owing to the amino acids encoded by exon 11. It would be intriguing to understand any structural differences between the two IR isoforms upon either insulin or S597 binding. Especially in light of the differential downstream signalling activation in different tissues that are associated with different expression of IR isoforms, IGF1R and consequently IR/IGF1R hybrid receptors.

Response: Given that insulin binding affinity for IR-A and IR-B is similar, we did not expect structural differences between insulin-bound IR-A and IR-B. Furthermore, the recent structural study using the IR-B isoform did not show the 12 amino acid extension in the C-terminal alpha subunit (PMID: 35660159), indicating that the 12-residues in IR-B might be very flexible. Because the a-CT motif of IR is not required for the S597-induced IR activation, we anticipate that S597 similarly activates IR-A and IR-B. Moreover, the *in vitro* functional assay demonstrated that S597 increases auto-phosphorylation of IR-A and IR-B, similar to insulin (**Fig. S6a,b**). We have included new data and expanded the discussion on the contribution of different expression of IR isoforms, IGF1R and IR/IGF1R heterodimer in the revised manuscript.

The insulin tolerance test data in Figure 5 g and h only extends to 60 minutes. Longer time courses such as 120 minutes are conventional and usually demonstrate a plateauing or rise of blood glucose at later time points. Were longer time-points collected? If not, why not? It would be important to understand whether blood glucose levels continue to fall at later time-points in S597 and S597-S20D treated mice.

Response: We thank the reviewer for this suggestion. The insulin tolerance assay was performed in 0, 15, 30, 60, 90, and 120 minutes following the injection of insulin (6 nmol/kg), S597 (9 nmol/kg), or S597-S20D (9 nmol/kg) (**Fig. 7b**). In mice treated with S597, glucose levels were significantly reduced and returned to normal 120 minutes after injection, similar to insulin. The glucose levels of mice treated with S597-S20D significantly decreased and remained stable over time, suggesting that S597-S20D is more effective in lowering blood glucose than S597. These results further suggest that agonist action of S597 can be modulated by modest sequence changes.

• Could the authors please address the following:

- Statistics are only provided for S597 analogue treated mice where blood glucose was elevated in response to administration. Presumably elevated blood glucose in S597 analogue treated mice (5g) is due to the partial binding of the binding site mutant analogues preventing endogenous insulin from binding and activating the IR, however this is not discussed in the manuscript.

Response: We thank the reviewer for these good suggestions. The manuscript has been revised accordingly.

- It is not clear as to what was used as the comparator for statistical significance. Please clarify this.

Response: We thank the reviewer for pointing out this. We clearly noted the comparator for statistical significance in the figures, figure legends, and source data.

- The figure suggests that insulin, S597 and S597-S20D administration did not statistically decrease blood glucose levels in these mice. Was this the case?

Response: In mice, insulin, S597, and S597-S20D administration significantly reduced blood glucose levels. We clearly noted the statistic results in the source data.

- Also, the strain of mice used for the in vivo studies is not stated in the manuscript. These details should be provided.

Response: We thank the reviewer for pointing out this mistake. In this study, we used C57BL/6J mice (stock #000664, Jackson Laboratory). This information has been noted in the section on methods.

The data demonstrate potential utility of S597 to activate insulin-binding deficient mutants. In light of the distinct IR structural changes that S597 binding causes, it would be important to examine the effects of S597 on IR mutants that perturb IR function in different ways, ie. mutations that occur away from the insulin binding site and perturb movement of hinge regions, and alter internalisation/recycling kinetics. This would demonstrate wider utility beyond mutations that affect insulin binding sites.

Response: We thank the reviewer for making this great suggestion. We have previously demonstrated that insulin-bound IR undergoes a large conformational change between the CR and L2 domains, and between L2 and FnIII-1 domains. These changes generate new intra- and inter-domain contacts, thus stabilizing compact T-shaped IR. Due to the distinct activation mechanism of IR upon S597 binding, we hypothesized that S597 activates IR mutants that disrupt stabilization of the compact T-shaped IR upon insulin binding. To test this, we examined the effects of insulin and S597 on IR mutants, R345A in the L2 domain and E697A in the N-terminal region of aCT, which form a salt bridge in the compact T-shape IR (note that both residues are located far from the bound insulin). Mutation of IR R345A or E697A greatly diminished IR activation by insulin, but those mutants could be fully activated by S597 (**Fig. 4e,f**). Indeed, these biochemical data are consistent with our cryo-EM results, which further confirm the structural observations of insulin-bound IR and S597-bound IR, suggesting that S597 and its analogs activate IR mutants that disrupt the insulin-binding site as well as maintaining the compact T-shaped IR stability via insulin binding. Additionally, we have included a sequence alignment of IR-A and IR-B from human, mouse IR-A, and human IGF1R to mark key residues (**Fig. S5**).

Minor Comments:

Inaccuracy in reporting of INSR mutants and cited literature. There are different substitutions for Arg14 described in ref 35 (R14W) and ref 20 (R14E). N15K is not described in ref 35. D496K is D496N in ref 36. Could this please be corrected.

Response: We thank the reviewer for pointing out this mistake. The correction has been made as follows. We cited literatures for IR N15K and corrected the patient mutants. In addition, we generated a patient-derived IR mutant, R14W, and demonstrated that IR R14W as well as IR R14A can be fully activated by S597, but not by insulin (**Fig. 4c-f**). We used IR D496K as the charge reversion mutation will significantly disrupt insulin binding compared to IR D496N.

Many of the figure legends throughout the manuscript do not state as to what the comparator was for the determination of statistical significance. Could this please be corrected.

Response: We thank the reviewer for pointing out this. In the figures, figure legends, and source data, we clearly indicate the comparator for statistical significance.

Figure 2g-j, concentrations and timescale of treatment stimulations of insulin and S597 are not indicated in the figure, the figure legend, or associated text. Please amend.

Response: Point accepted. We have made the correction.

Figure 2h, please update figure legend to indicate that KE/LA refers to K484E/L552A.

Response: Point accepted. We have made the correction.

Scale bar is not defined in extended data figure 6.

Response: Point accepted. We have made the correction.

Reviewer #3 (Remarks to the Author):

This study addresses S597, a single chain insulin receptor agonist developed around 20 years ago and studied intermittently since then. S597 has interesting signalling properties in cells, having been reported previously to favour metabolic over mitogenic signalling, and has also been shown to exert beneficial acute metabolic effects in mice.

This study moves understanding forward first by reporting use of cryoEM to solve a series of structures including the mouse insulin receptor and all or parts of S597, and second by providing evidence that the peptide can be used to activate naturally occurring mutated receptors, with potential implications for precision therapy for highly selected forms of insulin resistance caused by INSR mutations.

I found the paper generally well written and interesting. In places I thought the context in which the work is presented could be improved to make the study more accessible to general readers. I offer specific comments below. I am not a structural biologist, so I focus my comments in particular on signalling studies and studies of receptor mutations:

Response: We thank the reviewer for the positive assessment of our manuscript. We greatly appreciate the constructive comments which we have addressed below.

•Can the authors comment and justify why the short isoform of the insulin receptor was used for their studies? This is generally regarded as the less “metabolic” of the INSR isoforms, based in part on tissue expression patterns. Potential importance of INSR/IGF1R hybrids is involved in discussion of biased agonism, yet any differences in IR-A vs IR-B action could be just as relevant. Is there any structural reason to suppose differential effects of S597 on the two isoforms?

Response: We thank the reviewer for pointing out this. Due to the similar insulin binding affinity for IR-A and IR-B, we did not expect structural differences between insulin-bound IR-A and IR-B. Furthermore, the recent structural study using the IR-B isoform did not show the 12 amino acid

extension in the C-terminal alpha subunit (PMID: 35660159), indicating that the 12-residues in IR-B might be very flexible. In addition, as the a-CT motif is not required for the S597-induced IR activation, we anticipate that S597 similarly activates the IR-A and IR-B. To test this possibility, we expressed IR-A and IR-B in 293FT cells and examined the IR auto-phosphorylation upon insulin and S597 stimulation (**Fig. S6a,b**). We found that S597 increased auto-phosphorylation of IR-A and IR-B to a similar level as insulin. Nevertheless, we are planning to solve the structures of IR-B bound insulin or S597. We feel (and hope the reviewer would agree) that structural and functional analyses of IR-A and IR-B, and IR/IGF1R hybrid are beyond the scope of the current study.

•On a similar note, it is the mouse receptor that is used for structure determination, but cellular studies are undertaken using human INSR. This is clear from the methods but is not very explicit in results. Given high homology between species it is almost certain that findings between mouse/human are generalisable, but it may be worth commenting on conservation of key residues at the S597 binding sites.

Response: This is great suggestion. We have included alignments of human IR-A and IR-B, and mouse IR-A sequences and highlighted key residues including insulin site-1, S597, and L1/a-CT interactions in this revised manuscript (**Fig. S5**). In addition, we noted the sequence homology in the revised manuscript.

•The authors might consider adding a further cartoon or similar to Fig 3 to illustrate how the S597-bound structure is similar and different to the insulin-bound structure.

Response: We thank the reviewer for making this great suggestion. We have added insulin bound IR dimer in **Fig. 3e**.

•The study focuses on an informative group of INSR mutations that are said to have been found to be disease-causing in humans. However this is not completely clear from the references given. The human genetic reference 35 given for R14A seems to describe R14W (R41W) only on quick reading (doi:10.1007/s00592-013-0490-x) while I could not see N15K mentioned at all in that article. Descriptions of identification of D496K and D707A are clear. As new mutations continue to be discovered in recessive INSR defects I don't think it is critical that all mutations studied have been previously described in humans – this study is more about proof of concept. However could the authors please make sure their citation is accurate.

Response: We thank the reviewer for pointing out this mistake. The correction has been made as follows. We cited literatures for IR N15K and corrected the patient mutants. In addition, we generated a patient-derived IR mutant, R14W, and demonstrated that IR R14W as well as IR R14A can be fully activated by S597, but not by insulin (**Fig. 4e,f**). We used IR D496K as the charge reversion mutation will significantly disrupt insulin binding compared to D496N. Indeed, these data are consistent with our cryo-EM results, which further support the idea that S597 activates the IR in a distinct manner.

•Based on the structures they present, can the authors offer more explicit comment on what types of missense mutations would be amenable to activation by S597, and which wouldn't, in all likelihood? Obviously only those which are expressed at the cell surface rather than being misfolded and degraded will be accessible.

Response: We thank the reviewer for this great suggestion. As the reviewer pointed out, we do not anticipate the S597 could activate IR mutants that may cause misfolding, trafficking or kinase

activity. We speculate that S597 could activate IR mutants that disrupt normal insulin binding or stabilization of the compact T-shape of IR. We have added more discussion in the main text and prepare new **Fig. 4a,b** and **Fig. S5** to address this point.

•The paper perpetuates the long standing notion of a ‘metabolic’ pathway downstream from the INSR mediated by PI3K/AKT, and a “Mitogenic” pathway mediated by MAPK. However growth is really a metabolic phenomenon, the PI3K/AKT pathway is close to the most mutationally activated signalling pathway in cancer, and in any case there is clear crosstalk between RAS and PI3K, including direct binding. Is it time to downplay this simplistic dichotomisation?

Response: Point accepted. We have made the correction.

•The study shows that in some contexts S597 activates both PI3K/AKT and RAS/MEK/ERK signalling, but in others *in vivo* activation only of PI3K/AKT occurs. Very limited discussion of this focuses on a possible role of INSR/IGF1R heterodimers, which differ among tissues based on expression patterns of INSR and IGF1R. No data are shown to address this possibility, and as mentioned above IR-A/IR-B ratios could also be involved. Many other explanations are also possible, for example relating to the pharmacokinetics and signalling dynamics of S597 – what concentrations reach different interstitial spaces, and what are the effects on recycling dynamics etc on chronic exposure? This is speculation only, but at present I think the discussion is too simplistic and may be made more nuanced.

Response: We thank the reviewer for raising this important point. A limitation of our current *in vivo* study is that IR signaling in multiple tissues were examined at fixed time points and at a single dose. It is totally possible that pharmacokinetics and signaling dynamics of S597 may result in the selective IR signaling in different tissues. In addition, the expression levels of IR isoforms, IGF1R, and IR/IGF1R hybrid could contribute the IR signaling by S597. We have added more discussion in the main text.

•A previously reported approach to activating wild type and mutant insulin receptors, dating from the early 1990s, relies on bivalent ligands such as intact antibodies, whose stimulatory activity relies on bivalency and receptor cross linking. This can activate receptors in cells and induce some glucose lowering in animal models, including models of receptoropathy (e.g. PMID 32816962). One problem with that approach is that receptor downregulation results, severely blunting metabolic benefits. In this paper, brief acknowledgement of this longstanding, parallel line of work seeking to activate mutant antibodies may benefit readers. Do the authors have any evidence that chronic treatment with S597 is more or less likely than chronic insulin treatment to downregulate cell surface receptor? A monovalent single chain peptide such as S597 may well have distinct advantages over crude reagents that activate by cross linking, and it would be interesting to note this.

Response: We thank the reviewer for this great suggestion.

A previous study showed that S597 treatment of Zucker diabetic fatty rats using osmotic mini-pump for 7 days lowered glucose and increased triglyceride deposition in adipose tissue (PMID: 25315006). In addition, a recent study demonstrated that twice-daily injections of S597 into mice fed a diabetogenic diet for 18 weeks acutely reduced blood glucose and plasma triglyceride levels but did not lead to obesity or insulin resistance (PMID: 29483182). Neither study demonstrated a downregulation of the functional IR as compared with chronic insulin treatment. We also examined the chronic effects of S597 on the functional IR levels in two ways; (1) treatment with insulin or S597 at concentrations exceeding normal levels (50U/kg body weight) for three consecutive days

(Fig. S10a-e) and (2) one hour after the treatment with physiological levels of insulin (0.1U/kg) or S597 (0.15U/kg) (Fig. S10f). The IR was not significantly downregulated in either condition. We added new data and expanded discussion in the revised manuscript.

•It is assumed that there may be a specific structural basis for biased agonism, but similar biased agonism is seen for the bivalent antibodies mentioned above. This makes me wonder if there is a more generalisable, kinetic explanation for biased agonism induced by different types of ligand, to re-iterate one of the points made above?

Response: We thank the reviewer for raising this important point. We would like to point out that the mechanisms by which bivalent antibodies and S597 activate IR as well as their potencies to activate IR are significantly different.

The IR monoclonal antibodies such as 83-7 and 83-14 act as agonists for the IR. These Fab fragments of 83-7 and 83-14 were also used in determining the crystal structure of insulin bound IR (PMID: 23302862). The Fab 83-7 binds to L1 domain of IR and modestly enhances the insulin binding to IR (PMID: 2427071; PMID: 2832148). Meanwhile, the Fab 83-14 binds to FnIII-1 domain of IR and modestly inhibits the insulin binding. By combining insulin with antibodies, the IR auto-phosphorylation increases the maximal response to insulin by approximately 150%, whereas it can be increased by approximately 12% by treatment with those antibodies alone (PMID: 29700562). Interestingly, the IR monoclonal antibodies could increase the auto-phosphorylation of IR mutants including D707A by approximately 10% of the maximal IR WT response to insulin. While these antibodies alone could induce low levels of auto-phosphorylation in IR mutants, they were able to effectively increase pAKT, but not pERK. Similarly, the monoclonal antibody XMetA is an allosteric partial agonist of the IR (but not IGF1R), which also selectively activates PI3K-AKT pathway (PMID: 22403294; PMID: 23039274; PMID: 24876415 PMID: 25613982). Different from S597, this monoclonal antibody acts allosterically and does not compete with insulin for binding to the IR. It should also be noted that the maximal effect of XMetA on IR auto-phosphorylation is significantly less than that of insulin, which is only 20% of insulin's maximal effect. Based on these observations, Adams and colleagues conclude that XMetA's preferential signaling is caused by the inherent difference in pathway sensitivity between AKT and ERK responses to IR activation rather than a separate pathway-biased mechanism.

In cultured cells, S597 could activate the IR WT and even IR mutants (*e.g.* IR D707A) by approximately 80% maximal IR WT response to insulin, suggesting that the potency of S597 is higher than those monoclonal antibodies of IR. In this study, we demonstrated that the binding of S597 to apo-IR first delocalizes the α -CT from the L1 domain of IR by competing for the same binding surface on the L1 domain. This delocalization of the α -CT would lead to the destabilization of the auto-inhibitory conformation of IR. Subsequently, the two protomers could undergo rigid-body rotation using the L2-FnIII-1 interface as the hinge. Most importantly, S597 could stabilize the active state of IR by concurrently binding to the L1 domain of one protomer and the FnIII-1 domain of another. In the extended T-shaped, S597-bound IR reduces the distance between two FnIII-3 domains, resulting in two kinase domains in close proximity to undergo trans-autophosphorylation. We speculate that the IR monoclonal antibodies might destabilize the auto-inhibitory state of IR by intramolecular cross-linking but could not stabilize the active conformation, thereby it could partially activate the IR.

In mice, S597 activated IR by approximately 40% of the maximal response to insulin in liver and adipose tissue, but pAKT and pERK levels were comparable to insulin. Strikingly, S597 activated IR and pAKT in skeletal muscle similarly to insulin, while pERK was significantly impaired. It remains unclear how S597-bound IR selectively activates PI3K-AKT pathways in the

skeletal muscle. Given the fact that the biased signaling is observed in skeletal muscle but not in liver suggest that the relative expression of IR, IGF1R, and IR/IGF1R heterodimer contribute to the selectivity of IR signaling, as S597 specifically binds and activates IR, but not IGF1R. Indeed, healthy hepatocytes or mature adipocytes do not express appreciable levels of IGF1R and predominantly express IR, whereas skeletal muscle cells express both IR and IGF1R. Furthermore, although S597 did not change the IR auto-phosphorylation in cells overexpressing either IR-A or IR-B (**Fig. S6a,b**), we cannot rule out contributions from heterodimers formed between IR-A and IR-B (i.e. IR-A/IR-B), or even between IR isoforms and IGF1R (i.e. IR-A/IGF1R or IR-B/IGF1R) *in vivo*.

Because S597-S20D can activate both IR signaling pathways in skeletal muscle, we plan to determine the structure of IR/S597-S20D and compare it with the IR/S597. We believe that delineating the IR structural basis for differential activation of these two pathways would provide the molecular basis for that difference. We added discussion about these points and a limitation of our current *in vivo* study in the revised manuscript.

•In their summing up at the end of the discussion, the authors comment briefly that “As S597 or S597 analogs provide a significant advantage over a previously described insulin mimetic that showed low receptor specificity, it will be intriguing to ask whether insulin in combination with S597 could lower insulin requirements in patients with diabetes, decreasing the risk of hypoglycemia, weight gain, and oncogenesis associated with insulin therapy alone”, citing a 2012 review from when the debate about oncogenicity of currently used insulin was live. This simple and very speculative statement masks a great deal of clinical and epidemiological complexity and looks very alarming to the uninitiated regarding chronic insulin use. The authors do not elaborate on how S597 might have these beneficial effects. I suggest toning it down a little.

Response: Point accepted. We have toned down our statement.

In conclusion, we are grateful to the referees for their thoughtful and critical comments that have greatly improved our manuscript.

Reviewer #1 (Remarks to the Author):

I was already supporting the publication of this paper and I am happy to confirm my advice regarding publication given that the authors carefully answered all the points from reviewers #2 and #3.

Reviewer #2 (Remarks to the Author):

Title: Activation of the insulin receptor by an insulin mimetic peptide

This paper describes the receptor activation mechanism of the insulin mimetic peptide S597 through the description of the cryo-EM structure of S597 bound to the INSR-A. Furthermore, it examines the potential utility of S597 to activate mutations of the INSR. The work is novel, and INSR agonists that overcome defects in the INSR could be potentially lifesaving for those with rare syndromes of severe insulin resistance. The authors have addressed all the comments I made in my review of the original manuscript, and I recommend the revised version of the manuscript be accepted for publication.

Reviewer #3 (Remarks to the Author):

The authors have worked hard to respond to the reviewers' comments; The context of the studies undertaken is now well presented, and although uncertainties remain about the signalling effects of S597 on mutant receptors, these are reasonably acknowledged. This study provides useful new insights and in my opinion merits being read by the community.

REVIEWERS' COMMENTS

Reviewer #1 (Remarks to the Author):

I was already supporting the publication of this paper and I am happy to confirm my advice regarding publication given that the authors carefully answered all the points from reviewers #2 and #3.

Response: We thank the reviewer for the positive assessment of our manuscript.

Reviewer #2 (Remarks to the Author):

Title: Activation of the insulin receptor by an insulin mimetic peptide

This paper describes the receptor activation mechanism of the insulin mimetic peptide S597 through the description of the cryo-EM structure of S597 bound to the INSR-A. Furthermore, it examines the potential utility of S597 to activate mutations of the INSR. The work is novel, and INSR agonists that overcome defects in the INSR could be potentially lifesaving for those with rare syndromes of severe insulin resistance. The authors have addressed all the comments I made in my review of the original manuscript, and I recommend the revised version of the manuscript be accepted for publication.

Response: This is very clear and succinct summary of our work. We thank the reviewer for the positive assessment of our manuscript.

Reviewer #3 (Remarks to the Author):

The authors have worked hard to respond to the reviewers' comments; The context of the studies undertaken is now well presented, and although uncertainties remain about the signalling effects of S597 on mutant receptors, these are reasonably acknowledged. This study provides useful new insights and in my opinion merits being read by the community.

Response: We thank the reviewer for the positive assessment of our manuscript.